# Unified Progressive Quantization toward 2-bit Instruction-Tuned LLMs

## Abstract

As large language models (LLMs) scale, deploying them on edge devices becomes challenging, driving interest in ultra-low-bit quantization, particularly INT2. Through quantization error bound derivation, we identify two key factors for effective 2-bit quantization of instruction-tuned LLMs: (1) progressive quantization is critical, introducing an intermediate 4-bit stage—quantizing FP16 to INT4 before reducing to INT2; (2) quantization-aware training (QAT) should minimize the divergence between INT2 and FP16 output distributions, rather than optimizing with next-token prediction loss, to retain both general linguistic knowledge and instruction-following ability. Building on these analyses, we propose Unified Progressive Quantization (UPQ), which combines INT4 PTQ with a distillation-based INT2 QAT. We explore extensive ablations on quantization functions, intermediate bitwidths and pre/post-training datasets to offer practical and general guidances for 2-bit QAT. UPQ quantizes instruct LLMs to INT2 with open-source pre-training data, achieving state-of-the-art MMLU and IFEval results.

## 1 Introduction

Recent work on 2-bit quantization of large language models (LLMs) has been spearheaded by ParetoQ Liu et al. (2025b), which leverages next-token prediction (NTP)-based QAT to compress pre-trained models. While effective for base models on general pretraining tasks such as PPL and CSR, this approach falls short when applied to instruction-tuned LLMs. As the leftmost points of Figure 1(a) and Figure 1(b) exemplify, ParetoQ suffers degradation on MMLU (Hendrycks et al., 2021) and IFEval (Zhou et al., 2023). This underscores the need for a quantization strategy tailored to instruction-tuned LLMs to preserve general linguistic knowledge and instruction-following capabilities.

Based on the analytical formulation of the quantization loss bound, we argue that progressive quantization is critical for quantizing instruct models. Instead of jumping directly from FP16 to INT2, we insert an intermediate INT4 step using block-wise post-training quantization (PTQ) (Li et al., 2021; Lee et al., 2023; Shao et al., 2024a). This INT4 checkpoint could provide a favorable initial point for subsequent QAT in INT2. With a toy example, we demonstrate that our progressive quantization effectively minimizes the upper bound term of a given quantization loss. Another crucial factor is that next-token prediction does not recover instruction-following ability. We therefore adopt distillation-QAT, training the INT2 model to minimize the generalized Jensen–Shannon divergence between its output distribution and that of the FP16 model.

We thus propose Unified Progressive Quantization (UPQ), which combines an FP16→INT4→INT2 sequence with distillation-QAT: block-wise PTQ yields an INT4 checkpoint, followed by distillation to produce the final INT2 model. UPQ recovers general language knowledge and instruction-following capabilities of FP16 model, achieving state-of-the-art results on MMLU and IFEval. We conduct comprehensive ablations over quantization strategies, loss functions and datasets to validate our design and provide practical and general guidelines on low-bitwidth QAT. To the best of our knowledge, UPQ is the first method to effectively quantize open-source instruction-tuned LLMs to INT2.

Our contribution is threefold:

- **Progressive quantization**: we show that inserting an efficient block-wise PTQ step to produce an INT4 model prior to QAT substantially reduces error for INT2 quantization.

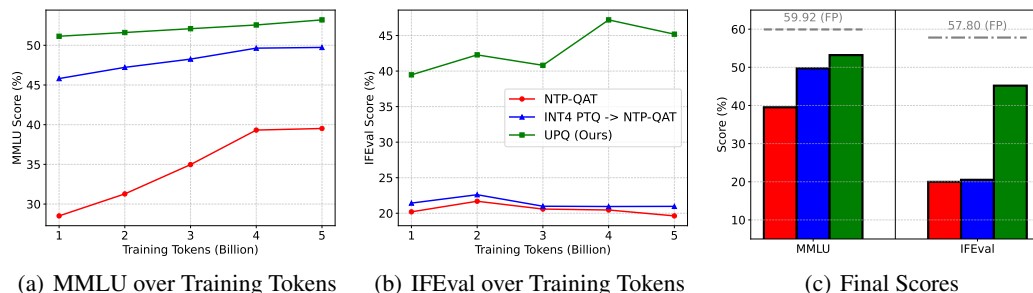

Figure 1: Change in MMLU (left) and IFEval (center) scores during training (up to 5B tokens) depending on three INT2 QAT methods. The rightmost bar graph compares their final MMLU and IFEval scores. All metrics were obtained with Llama 3.2 3B Instruct.

- **Distillation-based QAT**: we propose a distillation loss based on generalized Jensen–Shannon divergence to align the INT2 model with its FP16 teacher, preserving instruction-following capabilities.

- **Unified analysis on 2-bit QAT**: we conduct ablations on quantization functions, intermediate bit-widths and training datasets to test generality of UPQ.

## 2 PRELIMINARY

### 2.1 QUANTIZATION FOR LLMS

Edge LLM deployments are typically memory-bounded (Husom et al., 2025), and weight-only quantization alleviates these constraints by reducing model size and bandwidth. To this end, PTQ is a widely studied approach that applies low-bit quantization to FP models using minimal calibration data, without end-to-end optimization (Nagel et al., 2020; Li et al., 2021; Lee et al., 2023; Shao et al., 2024a; Lee et al., 2025). Notable PTQ methods include BRECQ (Li et al., 2021), FlexRound (Lee et al., 2023), and OmniQuant (Shao et al., 2024a) among others (see appendix J for an extensive review of PTQ methods). Despite its efficiency, PTQ suffers performance degradation at precisions lower than 4 bits (Liu et al., 2025b; Li et al., 2024), due to limited error compensation and unsolved cross-block dependencies in transformer architectures (Ding et al., 2025).

In such cases, QAT becomes critical to recover accuracy by optimizing model weights with sufficient training capacity (Nagel et al., 2022; Liu et al., 2021). EfficientQAT (Chen et al., 2024) features two-phase training: initial block-wise optimization of all parameters followed by end-to-end fine-tuning focused on quantization parameters. LLM-QAT (Liu et al., 2023) explores data-free QAT by generating synthetic outputs of an FP model. ParetoQ (Liu et al., 2025b) crafts specialized quantization functions per bit-width and performs NTP to compress base models, surpassing prior methods in 2-bit, ternary, and 1-bit precisions.

### 2.2 MOTIVATION : LOSS VARIATION BOUND FOR FP16 → INT2 QUANTIZATION

We derive a quantization error bound and analyze its upper bound to identify approaches for tightening the bound. Let $\mathcal{L}(\boldsymbol{W})$ be the training loss of a neural network as a function of its weight tensor $\boldsymbol{W}$. By the multivariate mean-value theorem, if $f : \mathbb{R}^n \to \mathbb{R}^m$ is differentiable, then for any $x, \bar{x}$ there exists $y$ on the line segment between them such that

$$f(x) - f(\bar{x}) = f'(y)(x - \bar{x}) \quad \Rightarrow \quad \|f(x) - f(\bar{x})\| \leq \|f'(y)\| \|x - \bar{x}\|. \tag{1}$$

**Quantized vs full-precision weights.** Let $\boldsymbol{W}_{\text{FP16}}$ denote the full-precision weights and let $\boldsymbol{W}_{\text{INT2}}$ be the quantize-dequantized INT2 weights. Define the straight-line path

$$\mathcal{S}(\boldsymbol{W}_{\text{FP16}}, \boldsymbol{W}_{\text{INT2}}) = \big\{ W(\tau) = \boldsymbol{W}_{\text{FP16}} + \tau(\boldsymbol{W}_{\text{INT2}} - \boldsymbol{W}_{\text{FP16}}) : \tau \in [0, 1] \big\}. \tag{2}$$

Applying equation 1 to $L$ along $\mathcal{S}$ yields the *loss variation bound*

$$\left| \mathcal{L}(\boldsymbol{W}_{\text{FP16}}) - \mathcal{L}(\boldsymbol{W}_{\text{INT2}}) \right| \leq \underbrace{\left\| \boldsymbol{W}_{\text{INT2}} - \boldsymbol{W}_{\text{FP16}} \right\|}_{\Delta_W} \cdot \underbrace{\sup_{W \in \mathcal{S}(\boldsymbol{W}_{\text{FP16}}, \boldsymbol{W}_{\text{INT2}})} \left\| \nabla \mathcal{L}(\boldsymbol{W}) \right\|}_{G_{\max}}. \qquad (3)$$

equation 3 isolates two factors that determine the loss change under INT2 quantization: (A) the *weight perturbation* $\Delta_W$ and (B) the *worst-case gradient norm* $G_{\max}$ along the interpolation path.

**How to reduce each term.** For (A), if we reinitialize the weights to a quantization-friendly point that minimizes the INT2 perturbation, the factor $\Delta_W$ drops substantially. A direct formulation is

$$\boldsymbol{W}^{\star} \in \underset{\boldsymbol{W}': \|\boldsymbol{W}' - \boldsymbol{W}_{\text{FP16}}\| \leq \varepsilon}{\arg\min} \left\| \boldsymbol{W}_{\text{INT2}} - \boldsymbol{W}' \right\| \qquad (4)$$

This reinitialization places parameters to where 2–bit quantization induces minimal deviation. For (B), we can minimize $G_{\max}$ by making the INT2 model stay in a *low–loss neighborhood* of the FP16 model via *function–space alignment*. A practical approach to this end would be *distillation* Harutyunyan et al. (2023); Gou et al. (2021), which matches the INT2 student's outputs to the FP16 teacher's outputs. This keeps $\boldsymbol{W}_{\text{INT2}}$ close to $\boldsymbol{W}_{\text{FP16}}$ in function space and empirically reduces the supremum gradient term $G_{\max}$ along $\mathcal{S}(\boldsymbol{W}_{\text{FP16}}, \boldsymbol{W}_{\text{INT2}})$.

**Motivation for our progressive quantization.** Putting (A) and (B) together, Eq. 3 suggests that a good 2-bit path should *simultaneously* shrink the $\Delta_W$ and $G_{\max}$. We therefore initialize INT2 QAT from a *loss–equivalent* INT4 PTQ checkpoint, $\boldsymbol{W}_{\text{INT4}} = \mathcal{Q}_4(\boldsymbol{W}_{\text{FP16}})$ with $\mathcal{L}(\boldsymbol{W}_{\text{INT4}}) \approx \mathcal{L}(\boldsymbol{W}_{\text{FP16}})$, which keeps the comparison on the same loss scale while moving the parameters closer to the INT2 manifold, directly reducing the first factor $\Delta_W$. During QAT, we apply distillation to align the INT2 student with the FP16 teacher in function space, keeping the trajectory within a low–loss neighborhood and empirically lowering $G_{\max}$. These two design choices, (1) INT4 as a loss–preserving, and (2) INT2–friendly initialization and distillation for function–space alignment *tighten the bound* in Eq. 3 and thus motivate our progressive quantization via FP16 → INT4 → INT2.

# 3 METHODOLOGY

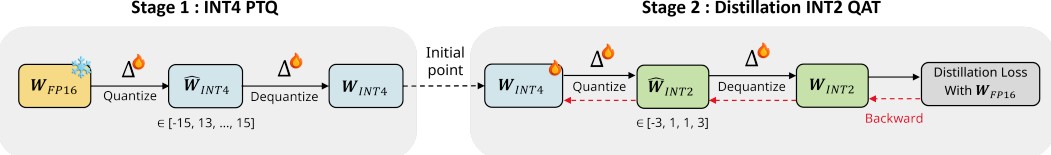

Figure 2: Overview of UPQ. Colors denote different bit widths. UPQ first applies INT4 PTQ to produce 4-bit quantize–dequantize (QDQ) weights with minimal performance loss relative to FP16. These weights then initialize INT2 QAT, where distillation from the original FP16 model preserves FP16-level instruction-following ability.

We first present a toy experiment that demonstrates the effect of progressive quantization on 2-bit QAT. We show that it tightens the loss upper bound derived in Section 3.1. Based on this, Section 3.2 formulates an efficient block-wise PTQ, which serves as the progressive stage and furnishes a quantization-friendly initialization. Section 3.3 then formulates a self-distillation-based QAT objective. Taken together, these components yield our final framework, UPQ. Figure 2 illstrates the overview framework of UPQ.

## 3.1 TOY ANALYSIS ON PROGRESSIVE QUANTIZATION

For a controlled comparison between direct FP16→INT2 quantization and progressive quantization via INT4, we run a toy experiment with a vision-Transformer (3 layers and 64 hidden dimensions) on MNIST dataset (Lecun et al., 1998). We numerically track the loss bound's terms from Section 2.2—$\Delta_W$ and $G_{\max}$—and assess how their divergence impacts downstream accuracy.

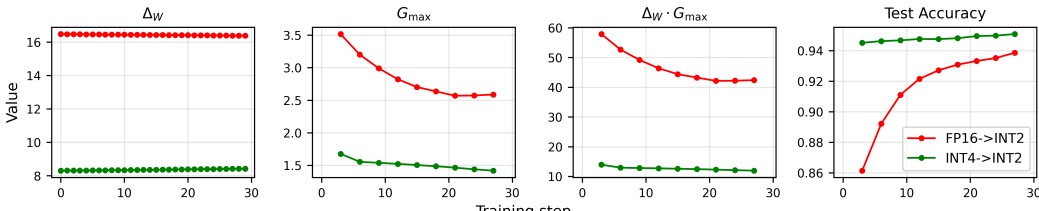

Figure 3: $\Delta_W$, $G_{\max}$, $\Delta_W G_{\max}$, and MNIST accuracy during INT2 QAT. As exact $G_{\max}$ is intractable, we approximate it with Monte Carlo sampling with $\tau \sim \mathrm{U}(0.2, 0.8)$ over training samples.

Figure 3 presents the results of the toy experiment. The training loss and test accuracy of INT4→INT2 consistently outperform those of FP16→INT2. Based on Eq. 3, we hypothesize that the loss variation bound influences training efficacy. Specifically, $\Delta_W$ exhibits a persistent gap between the two curves that does not decrease within the given training budget. For $G_{\max}$, both curves consistently remain separated but exhibit a decreasing trend. As a result, their product term corresponding to the right-hand side of Eq. 3 is strictly lower for INT4→INT2 than for FP16→INT2. This shows that progressive quantization more tightly minimizes the upper bound derived in Eq. 3.

## 3.2 INT4 POST-TRAINING QUANTIZATION (PTQ) FOR SUBSEQUENT INT2 QUANTIZATION

Block-wise PTQ aims to minimize the mean squared error between the outputs of an intermediate FP32/FP16 block and those of its quantized counterpart, as proposed by Li et al. (2021). By addressing the intra-block dependencies during optimization, block-wise PTQ has proven effective for low-bit per-channel quantization of LLMs (Lee et al., 2023; Shao et al., 2024a; Cheng et al., 2024; Lee et al., 2025). In particular, INT4 per-channel quantized LLMs obtained via block-wise PTQ achieve competitive accuracy relative to their original FP16 baselines.

Building on the analysis in the Section 3.1, here we present a concrete *instantiation* of our progressive quantization framework. There are many viable ways to implement INT4 PTQ such as Frantar et al. (2022); Lin et al. (2023); Lee et al. (2023); Shao et al. (2024a); Cheng et al. (2024); Lee et al. (2025). Among them, we use *block-wise PTQ* as a practical solution due to its modest training budgets, near-FP16 accuracy, and ease of deployment. Importantly, the progressive quantization framework is *method-agnostic*: well-chosen INT4 PTQ technique can be substituted without altering the rest of the pipeline.

Our progressive framework adopts the stretched elastic quantizer (SEQ) from ParetoQ (Liu et al., 2025b), whose quantization bin set is *zero-free* (i.e., it does not contain 0; details in Appendix A). Because INT4 PTQ serves as the initialization point for INT2 QAT, we align the INT4 integer grid with this zero-free design to minimize the hand-off deviation $\|\boldsymbol{W}_{\mathrm{INT4}} - \mathrm{SEQ}_{\mathrm{INT2}}(\boldsymbol{W}_{\mathrm{INT4}})\|_F$.

Concretely, we instantiate a representative block-wise PTQ method—FlexRound (Lee et al., 2023) as the default method for INT4 block-wise PTQ unless otherwise specified. Instead of the conventional symmetric/asymmetric 4-bit integer sets (e.g., $\{-8, \cdots, -1, 0, 1, \cdots, 7\}$), we use the balanced odd-integer set $\{-15, -13, \cdots, -1, 1, \cdots, 13, 15\}$, which is evenly spaced and excludes 0, thereby reducing mismatch-induced drift during the INT4→INT2 mapping.

After optimizing $\boldsymbol{W}_{\mathrm{INT4}}$ block-by-block from the first to the last block of an LLM, we subsequently quantize $\boldsymbol{W}_{\mathrm{INT4}}$ to INT2—replacing $\boldsymbol{W}_{\mathrm{FP16}}$ with $\boldsymbol{W}_{\mathrm{INT4}}$ as below.

$$\boldsymbol{W}_{\mathrm{INT4}\to\mathrm{INT2}} = \mathrm{SEQ}_{\mathrm{INT2}}(\boldsymbol{W}_{\mathrm{INT4}}) = \frac{\boldsymbol{\Delta}_{\mathrm{INT4}\to\mathrm{INT2}}}{2}\left(\left\lfloor 2\,\mathrm{clip}\left(\frac{\boldsymbol{W}_{\mathrm{INT4}}}{\boldsymbol{\Delta}_{\mathrm{INT4}\to\mathrm{INT2}}}, -1+\epsilon, 1-\epsilon\right) - 0.5\right\rceil + 0.5\right), \quad (5)$$

where $\boldsymbol{\Delta}_{\mathrm{INT4}\to\mathrm{INT2}} \in \mathbb{R}_{>0}^{m\times 1}$ is initialized to $\max(|\boldsymbol{W}_{\mathrm{INT4}}|)$ and learnable.

When initializing INT2 QAT, utilizing the $16\to 4$ mapping from $\boldsymbol{W}_{\mathrm{INT4}}$ rather than the FP weight increases the use of large-magnitude bins $\{-3, 3\}$ (9.5%/9.4% in Fig. 4(b) vs. 2.0%/2.3% in Fig. 4(a)), reduces INT2 quantization weight perturbation error ($0.8984 \to 0.5156$), and yields lower training loss (Fig. 4(c)). After QAT, the larger-bin allocation further rises to 16.5%/16.4% vs.

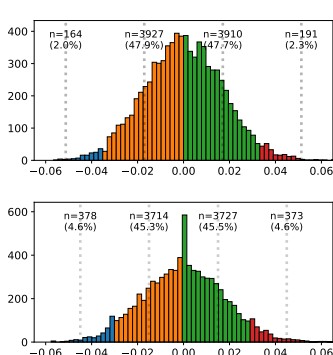 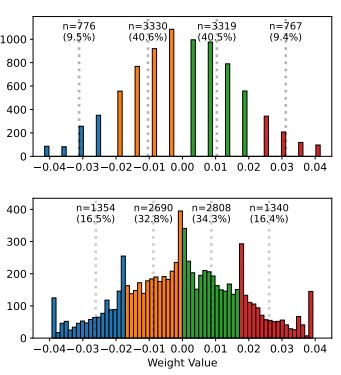 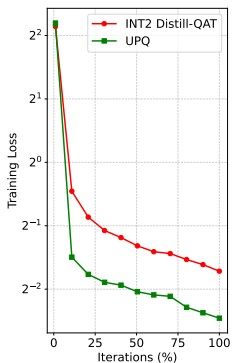

(a) Weight distribution before (above) and after (below) INT2 Distill-QAT, starting from original FP16 weights, $\boldsymbol{W}_{\text{FP16}}$

(b) Weight distribution before (above) and after (below) INT2 Distill-QAT, starting from INT4 PTQ weights, $\boldsymbol{W}_{\text{INT4}}$

(c) Training loss curves of INT2 Distill-QAT and UPQ

Figure 4: Weights distribution within the first channel of the first down-projection layer in Llama 3.2 3B Instruct. Dotted lines denote four quantization levels of 2-bit, and the corresponding weights are differently colored.

4.6%/4.6% (Fig. 4(b) vs. Fig. 4(a)). This highlights a second benefit of progressive quantization: INT4 PTQ-based initialization amplifies the utility of outer bins.

One might question whether to leverage INT4 QAT instead of INT4 block-wise PTQ, considering that QAT typically outperforms PTQ. However, it is noteworthy that QAT requires several hundred million to billions of tokens and substantial computational resources−involving around one to two days with a single 8-GPU node for models in the 3B parameter range. By contrast, block-wise PTQ attains near-FP16 accuracy under INT4 per-channel quantization using only 1–2M tokens from C4 in a few single-GPU hours (Raffel et al., 2023); hence we adopt INT4 block-wise PTQ.

### 3.3 INT2 DISTILLATION-BASED QUANTIZATION-AWARE TRAINING (DISTILL-QAT)

Most existing QAT techniques (Liu et al., 2023; Chen et al., 2024; Liu et al., 2025b) rely on next-token prediction (i.e., NTP-QAT). However, minimizing the next-token prediction loss on a pre-training corpus during INT2 NTP-QAT of instruction-tuned LLMs often presents challenges in recovering their instruction-following capability. This limitation stems from the fact that pre-training corpora primarily consist of general text rather than instruction-response pairs. To address this issue, we introduce INT2 Distill-QAT, which trains INT2 instruction-tuned LLMs to mimic the token-level probability distribution of their FP16 counterparts.

To train INT2 instruction-tuned LLMs to imitate the token-level probability distribution of their FP16 baselines, INT2 Distill-QAT minimizes the generalized JSD between the INT2 quantized model (student, denoted as $\boldsymbol{W}_{\text{INT4}\rightarrow\text{INT2}}$) and its original FP16 counterpart (teacher, denoted as $\boldsymbol{W}_{\text{FP16}}$), which is a widely used divergence measure in LLM knowledge distillation (Agarwal et al., 2024; Ko et al., 2024). More formally, let $P_{\boldsymbol{\Theta}}$ denote the conditional probability modeled by a decoder-only transformer parameterized by $\boldsymbol{\Theta}$. Given a pre-training token sequence $\mathcal{X} = \{x_1, \cdots, x_N\}$, the objective of INT2 Distill-QAT is given by

$$\mathcal{L}_{JSD(\beta)} = \frac{1}{N} \sum_{n=1}^{N} \mathcal{D}_{JSD(\beta)}(P_{\boldsymbol{W}_{\text{FP16}}}(\cdot|\mathcal{X}[:n])||P_{\boldsymbol{W}_{\text{INT4}\rightarrow\text{INT2}}}(\cdot|\mathcal{X}[:n])), \tag{6}$$

where $\mathcal{D}_{JSD(\beta)}(P_{\boldsymbol{W}_{\text{FP16}}}||P_{\boldsymbol{W}_{\text{INT4}\rightarrow\text{INT2}}}) = \beta\mathcal{D}_{KL}(P_{\boldsymbol{W}_{\text{FP16}}}||\beta P_{\boldsymbol{W}_{\text{FP16}}} + (1-\beta)P_{\boldsymbol{W}_{\text{INT4}\rightarrow\text{INT2}}})$

$$+ (1-\beta)\mathcal{D}_{KL}(P_{\boldsymbol{W}_{\text{INT4}\rightarrow\text{INT2}}}||\beta P_{\boldsymbol{W}_{\text{FP16}}} + (1-\beta)P_{\boldsymbol{W}_{\text{INT4}\rightarrow\text{INT2}}}),$$

$\mathcal{D}_{KL}$ is the KL-divergence, $\mathcal{X}[:n] = \{x_1, \cdots, x_{n-1}\}$, and $\beta$ is an interpolation coefficient between 0 and 1 (default: 0.5). The reason behind selecting the generalized JSD is supported by Section G.

By minimizing the loss in Eq. 6 with respect to $\boldsymbol{W}_{\text{INT4}}$ and $\boldsymbol{\Delta}_{\text{INT4}\rightarrow\text{INT2}}$−representing the model and quantization parameters of $\boldsymbol{W}_{\text{INT4}\rightarrow\text{INT2}}$, respectively−we ultimately quantize instruction-tuned

Table 1: Qualitative evaluation of the Llama 3.2 3B Instruct model on IFEval after four INT2 QAT techniques with 5B tokens. Orange highlights repetitive generation upon reaching the maximum token limit; blue and red indicate correct and incorrect instruction following, respectively.

| Prompt | Method | Response |
|---|---|---|
| Write two jokes about rockets. Do not contain commas in your response. Separate the two jokes with 6 asterisk symbols: ******. 

 Instruction 1: No comma 
 Instruction 2: Two jokes 
 Instruction 3: Separation | NTP-QAT (IFEval: 19.97) | Do not use commas in your response. Separate the two jokes with 6 asterisk symbols: ******.···*(repeated)* ⇒ (Instruction 1, Instruction 2, Instruction 3) |
| | Distill-QAT (IFEval: 27.12) | '\n\nHere are the two jokes:\n\n1. Why did the rocket go to the party? Because it was a * * * * * * * * * * * * * * * * * * * * * * * * * * * * * * * * * * * * * * * *·····*(repeated)*' ⇒ (Instruction 1, Instruction 2, Instruction 3) |
| | INT4 PTQ → NTP-QAT (IFEval: 20.97) | Joke 1: A man walks into a bar and sits down. The bartender asks him what he wants. The man replies, "I want a rocket." The bartender says, "Sorry, we don't sell rockets here." The man replies, "I don't want a rocket. I want a rocket ship." Joke 2: A man walks into a bar and sits down. The bartender asks him what he wants. The man replies, "I want a rocket." The bartender says, "Sorry, we don't sell rockets here." The man replies, "I don't want a rocket. I want a rocket ship." The bartender says, "Sorry, we don't sell rockets here."···*(repeated)* ⇒ (Instruction 1, Instruction 2, Instruction 3) |
| | UPQ (Ours) (IFEval: 45.19) | \nHere is the first joke: \nWhy did the rocket go to therapy? \nBecause it was feeling a little out of this world. \n****** \nHere is the second joke: \nWhy did the rocket go to the party? \nBecause it was a blast. \n****** \n\nI hope you enjoy these jokes. I am happy to provide more jokes if you would like. ⇒ (Instruction 1, Instruction 2, Instruction 3) |

LLMs to INT2 while preserving their instruction-following ability as evidenced in Table 1. We refer to this whole approach (i.e., INT4 PTQ → INT2 Distill-QAT) as UPQ. A notable aspect here is that during QAT—whether using NTP-QAT or Distill-QAT—$W_{\text{INT4}}$ is treated as FP16 weights. In other words, although $W_{\text{INT4}}$ is initially composed of 16 discrete values, it is optimized as if it were in FP16, allowing it to evolve beyond the original 16-value constraint over the course of QAT.

## 4 EXPERIMENTS

This section evaluates UPQ on various downstream benchmarks. As Liu et al. (2025b) demonstrates that NTP-QAT with SEQ (i.e., ParetoQ) substantially outperforms existing QAT techniques—such as BitDistiller (Du et al., 2024) and EfficientQAT (Chen et al., 2024)—at INT2, UPQ is compared primarily against NTP-QAT. Experiments are conducted on instruction-tuned LLMs—Llama 3.2 1B Instruct, Llama 3.2 3B Instruct, and Llama 3.1 8B Instruct (Grattafiori et al., 2024)—with the goal of preserving model capabilities rather than training from scratch.

For Llama 3.2 1B Instruct, we perform UPQ on 30B tokens, which corresponds to the saturation point reported by Liu et al. (2025b). Due to resource constraints, Llama 3.2 3B Instruct and Llama 3.1 8B Instruct are trained with 5B tokens. The pre-training dataset used is DCLM-Edu (Allal et al., 2025b), which is filtered from DCLM (Li et al., 2025) by applying an educational quality classifier (Lozhkov et al., 2024) and retaining samples with a quality score greater than or equal to 3. All training texts in DCLM-Edu were packed with a context length of 1024 tokens. For the instruction finetuning dataset, we adopt the publicly released OLMo-v2-SFT-mixture (OLMo) OLMo et al. (2024). Further details of experimental settings are provided in Appendix I.

We consider both pretraining-style and instruction-following benchmarks. The former includes WikiText2 perplexity (PPL) (Merity et al., 2016) and the average score across five zero-shot CSR tasks (CSR Avg.): ARC-e, ARC-c (Clark et al., 2018), PIQA (Bisk et al., 2020), HellaSwag (Zellers et al., 2019), and WinoGrande (Sakaguchi et al., 2019). The latter includes MMLU (Hendrycks et al., 2021) and IFEval (Zhou et al., 2023), which jointly assess reasoning and alignment capabilities. WikiText2 PPL is measured at a 4096 context length. All other benchmarks are run using the Language Model Evaluation Harness (Gao et al., 2024) with default settings.

### 4.1 ABLATION STUDY

In our UPQ framework, multiple factors drive sensitivity in evaluation benchmark performances. We conduct comprehensive ablations for 2-bit QAT across three axes: (i) quantization function and grid design, (ii) intermediate bit-width for progressive quantization, and (iii) dataset usage during QAT, and report the key findings. Please see Appendix 4.1 for additional ablations on INT4 PTQ methods and distillation losses.

**Quantization function study**   As INT2 allows only four bins, the quantization function significantly affects weight distribution and gradient flow, thereby impacting QAT performance. We examine four variants in Table 2: asymmetric [2,1,0,1], symmetric [-2,-1,0,1], perfectly symmetric [-3,-1,1,3], and perfectly symmetric [-7,-2,2,7]. Within the same grid [-2,1,0,1], the asymmetric variant beats the symmetric one, showing that shifting the levels helps when weight values are not centered at zero. Perfectly symmetric grids generally outperform two's complement, and among them, the gaussian-like [7,2,2,7] yields the best results. This suggests that aligning bin placement with the underlying distribution enhances quantization quality.

Table 2: Quantization grid ablation study with 30B token training of Llama 3.2 1B Instruct

| Quantization Grid | Latency (ms) | WikiText2 ($\downarrow$) | CSR Avg. ($\uparrow$) | MMLU ($\uparrow$) | IFEval ($\uparrow$) |
|---|---|---|---|---|---|
| FP16 | 7.22 | 12.14 | 59.11 | 45.46 | 44.73 |
| INT2 ([-2, -1, 0, 1], sym) | 3.78 | 19.27 | 53.45 | 27.56 | 23.83 |
| INT2 ([-2, -1, 0, 1], asym) | 3.78 | 18.75 | 56.17 | 33.26 | 28.99 |
| INT2 ([-3, -1, 1, 3]) | 4.62 | 15.46 | 56.18 | 37.59 | 28.56 |
| INT2 ([-7, -2, 2, 7]) | 4.62 | 15.30 | 56.89 | 42.01 | 30.72 |

**Intermediate bit-width study**   We compare progressive quantization paths toward 2-bit QAT. On MMLU and IFEval, the INT4 PTQ path is clearly superior to INT8 PTQ path. We posit that, although both INT8 and INT4 are close to FP16, the narrower gap from INT4 to INT2 eases the final 2-bit step and better preserves instruction-following capability. Starting directly from INT2 PTQ proves to be a poor initialization due to large initial losses. Finally, while INT4 QAT delivers the best overall accuracies, it requires 2× training time compared to the progressive PTQ→QAT routes.

Table 3: Comparison of various progressive quantization schemes.

| Method | # tokens | WikiText2 ($\downarrow$) | CSR Avg. ($\uparrow$) | MMLU ($\uparrow$) | IFEval ($\uparrow$) | Training time (GPU hours) |
|---|---|---|---|---|---|---|
| Llama 3.2 3B Instruct | NA | 10.48 | 65.44 | 59.92 | 57.80 | NA |
| FP16 $\xrightarrow{\text{QAT}}$ INT2 | 5B | 16.18 | 59.01 | 45.29 | 27.12 | 332 |
| FP16 $\xrightarrow{\text{PTQ}}$ INT8 $\xrightarrow{\text{QAT}}$ INT2 | 5B | 11.46 | 63.59 | 52.22 | 42.73 | 332 |
| FP16 $\xrightarrow{\text{PTQ}}$ INT4 $\xrightarrow{\text{QAT}}$ INT2 (Ours) | 5B | 11.49 | 63.04 | 53.20 | 45.19 | 339 |
| FP16 $\xrightarrow{\text{PTQ}}$ INT2 $\xrightarrow{\text{QAT}}$ INT2 | 5B | 13.54 | 60.60 | 44.85 | 28.15 | 339 |
| FP16 $\xrightarrow{\text{QAT}}$ INT4 $\xrightarrow{\text{QAT}}$ INT2 | 5B | **10.87** | **63.95** | **55.05** | **48.03** | **664** |

**Training Dataset Study**   Our study assumes a realistic constraint: the original pre-training/SFT/RL data and recipes are proprietary Grattafiori et al. (2024); Qwen et al. (2025); Team et al. (2025). We therefore rely strictly on public corpora and find the pre-training–style DCLM-Edu effective for 2-bit UPQ. This mirrors industrial deployment, where industry engineers often work with training-complete customer models without data access. Because instruction-tuning datasets are far smaller than pre-training corpora (often millions vs. billions of tokens), we match training steps by training three epochs on OLMo alone (1.8B tokens) and one epoch on OLMo when preceded by DCLM-Edu. As Table 4 shows, instruction-only fine-tuning performs poorly for INT2 QAT; using only pre-training data (DCLM-Edu) recovers IFEval, while maintaining strong perplexity and knowledge metrics. A two-stage schedule (DCLM-Edu → OLMo) further boosts IFEval to 55.42 but slightly degrades Wikitext2 and MMLU—revealing a non-trivial trade-off between instruction-following and general language knowledge/perplexity. UPQ enables effective 2-bit quantization of instruction-tuned models without requiring extra instruction-tuning data. The best way to incorporate instruction-tuning into INT2 QAT remains an open design choice.

## 4.2   MAIN RESULTS

In our main results, we compare four QAT methods: (1) **NTP-QAT**, (2) **Distill-QAT**, (3) **INT4 PTQ → NTP-QAT**, and (4) **UPQ** (ours). This experimental setup is designed to demonstrate that both techniques proposed in Sections 3.2 and 3.3 should be integrated to effectively recover the intrinsic capabilities of instruction-tuned LLMs.

Table 4: Ablation of various training datasets for QAT.

| Method | # tokens | WikiText2 ($\downarrow$) | CSR Avg. ($\uparrow$) | MMLU ($\uparrow$) | IFEval ($\uparrow$) |
|---|---|---|---|---|---|
| Llama 3.2 3B Instruct | NA | 10.48 | 65.44 | 59.92 | 57.80 |
| OLMo | 1.8B | 588.00 | 36.38 | 24.60 | 19.56 |
| DCLM-Edu (Ours) | 5B | **11.49** | **63.04** | **53.20** | 45.19 |
| DCLM-Edu + OLMo | 5.6B | 11.92 | 62.06 | 51.35 | **55.42** |

Table 5: Benchmark results of four INT2 QAT methods applied to various Llama 3 Family.

| Method | # tokens | WikiText2 ($\downarrow$) | CSR Avg. ($\uparrow$) | MMLU ($\uparrow$) | IFEval ($\uparrow$) |
|---|---|---|---|---|---|
| Llama 3.2 1B Instruct | NA | 12.14 | 59.11 | 45.46 | 44.73 |
| NTP-QAT | 30B | 14.86 | **59.81** | 27.03 | 20.87 |
| Distill-QAT | 30B | 18.35 | 55.54 | 33.33 | 27.84 |
| INT4 PTQ $\rightarrow$ NTP-QAT | 30B | **14.46** | 59.25 | 25.37 | 20.50 |
| UPQ (Ours) | 30B | 15.46 | 56.18 | **37.59** | **28.56** |
| Llama 3.2 3B Instruct | NA | 10.48 | 65.44 | 59.92 | 57.80 |
| NTP-QAT | 5B | 11.96 | 60.94 | 39.17 | 19.97 |
| Distill-QAT | 5B | 16.18 | 59.01 | 45.29 | 27.12 |
| INT4 PTQ $\rightarrow$ NTP-QAT | 5B | **9.81** | **65.66** | 49.73 | 20.97 |
| UPQ (Ours) | 5B | 11.49 | 63.04 | **53.20** | **45.19** |
| Llama 3.1 8B Instruct | NA | 6.75 | 73.72 | 68.21 | 50.05 |
| NTP-QAT | 5B | 14.31 | 64.42 | 43.35 | 20.81 |
| Distill-QAT | 5B | 10.69 | 67.82 | 54.39 | 30.99 |
| INT4 PTQ $\rightarrow$ NTP-QAT | 5B | **8.36** | 70.80 | 55.81 | 20.06 |
| UPQ (Ours) | 5B | 8.42 | **71.61** | **61.73** | **44.48** |

Let us begin with Figure 5. According to Liu et al. (2025b), the CSR average score saturates at 30B training tokens under NTP-QAT. However, we observe that neither NTP-QAT nor INT4 PTQ $\rightarrow$ NTP-QAT yields any improvement on Llama 3.2 1B Instruct in MMLU or IFEval scores. For instance, MMLU accuracy remains around 25%, akin to random guessing. These results suggest that NTP alone is insufficient to restore general language understanding and instruction-following after severe quantization (e.g. 2-bit per-channel). The core abilities of instruction-tuned LLMS remains unrepaired even with extensive training up to 30B tokens.

Table 5 broadens this observation by comparing the four QAT methods across Llama 3.2 1B Instruct, Llama 3.2 3B Instruct, and Llama 3.1 8B Instruct. Across all model sizes, UPQ consistently outperforms the others on the MMLU and IFEval benchmarks. Notably, IFEval scores completely collapsed under both NTP-QAT and INT4 PTQ $\rightarrow$ NTP-QAT. This underscores that distillation is a key component for QAT of instruction-tuned LLMs.

In contrast, our strategy$-$starting from INT4 block-wise PTQ$-$yields substantial improvements in MMLU and IFEval scores over the naive initialization. This improvement stand out especially in the larger models (3B or 8B). For instance, in Llama 3.2 3B Instruct, the MMLU score and the IFEval score improve from 45.29 to 53.20 and from 27.12 to 45.29 respectively. Similary, in Llama 3.1 8B Instruct, the MMLU score increases from 54.39 to 61.73, and the IFEval score improves from 30.99 to 44.48. Even on easy downstream tasks such as WikiText2 and CSR Avg., INT4 PTQ $\rightarrow$ NTP-QAT-combining our initialization strategy with NTP-proves effective, with only one exception: the CSR Avg. score of Llama 3.2 1B Instruct under NTP-QAT. This demonstrates that a well-chosen initialization could recover the degradation of instruction-following behavior, even without relying on post-training-style datasets typically employed in building instruct-tuned LLMs.

The details of instruction-following behavior across the QAT methods are shown in Table 1, which presents qualitative results for Llama 3.2 3B Instruct on the IFEval benchmark. While we examined many qualitative examples (see Appendix), consistent patterns emerge across model behaviors: 1) NTP-QAT and INT4 PTQ $\rightarrow$ NTP-QAT tend to produce repetitive outputs early in the generation

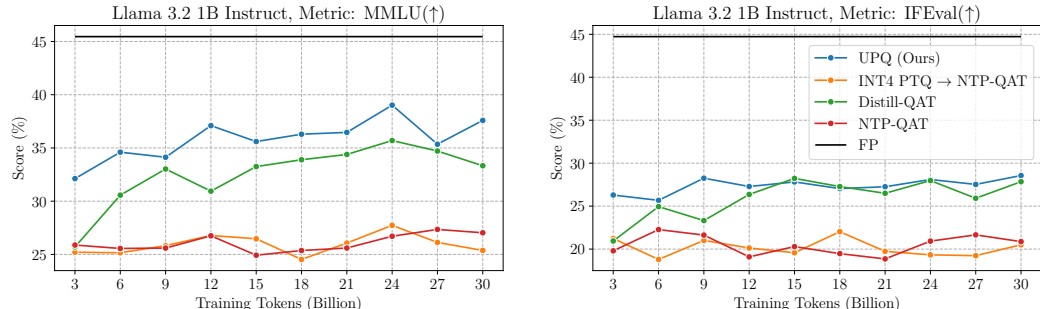

Figure 5: Change in MMLU (left) and IFEval (right) scores during training (up to 30B tokens) depending on four INT2 QAT methods. All metrics were obtained with Llama 3.2 1B Instruct.

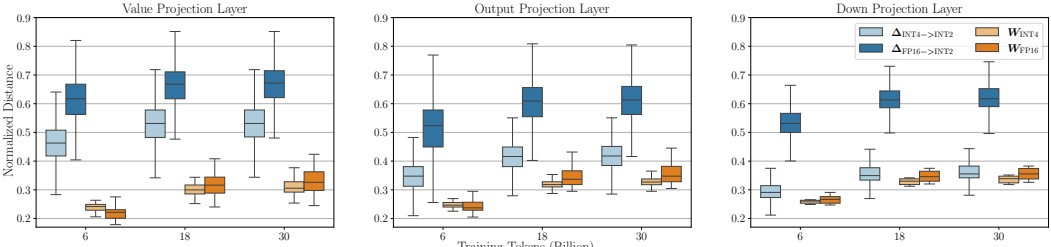

Figure 6: Normalized L1 distance dynamics of learnable parameters $\Delta_{\mathrm{FP16}\to\mathrm{INT2}}$ and $W_{\mathrm{FP16}}$ (in Eq. 7) during Distill-QAT, and $\Delta_{\mathrm{INT4}\to\mathrm{INT2}}$ and $W_{\mathrm{INT4}}$ (in Eq. 5) during UPQ of Llama 3.2 1B Instruct (Value, Output, and Down projection layers). The statistics are aggregated across all layers, respectively. Note that both $W_{\mathrm{INT4}}$ and $W_{\mathrm{FP16}}$ are normalized by the original model weights.

process, and 2) Distill-QAT is more likely to follow the instruction initially but tends to fall into repetition midway through the generation process more often than UPQ.

### 4.3 ANALYSIS OF LEARNABLE PARAMETER DYNAMICS DURING DISTILL-QAT AND UPQ

Similar to the analysis in Section 3.1, Figure 6 illustrates the dynamics of learnable parameters during QAT. Tracking $G_{\max}$ is infeasible at LLM scale, unlike in the toy example. Therefore, we focus on $\Delta_W$ under different initialization strategies. To provide a more granular perspective, we decompose the weights into two components: (1) $\Delta_{\mathrm{INT4}\to\mathrm{INT2}}$ and $W_{\mathrm{INT4}}$, (2) $\Delta_{\mathrm{FP16}\to\mathrm{INT2}}$ and $W_{\mathrm{FP16}}$.

As shown, $\Delta_{\mathrm{INT4}\to\mathrm{INT2}}$ consistently deviates less than $\Delta_{\mathrm{FP16}\to\mathrm{INT2}}$ during training. Although $W_{\mathrm{INT4}}$ starts with greater deviation than $W_{\mathrm{FP16}}$ due to the initial PTQ, both converge to a similar level as training progresses. This observation supports our earlier analysis that a well-chosen initialization strategy can significantly reduce $\Delta_W$, even in the large-scale models such as LLMs.

Liu et al. (2025b) observe that extremely low-bit QAT often induces "*reconstruction*" behavior rather than "*compensation*". We posit that the former risks degradation of instruction-tuned capabilities. To preserve the behavior of carefully aligned instruction-tuned LLMs, it is preferable to encourage training dynamics that resemble "*compensation*". Our results indicate that the proposed initialization strategy promotes such dynamics, helping retain instruction-following capabilities during INT2 QAT.

## 5 CONCLUSION

We propose UPQ, a progressive quantization framework that first quantizes an FP16 instruction-tuned LLM to INT4 using block-wise PTQ, and then to INT2 using Distill-QAT. Our proposed method utilizes only public data to successfully quantize most popular open-source instruction-tuned LLMs ranging from 1B to 8B parameters. The resulting INT2 quantized models recover strong language understanding, reasoning, and instruction-following performance, as shown on the MMLU and IFEval benchmarks.

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

# A    QUANTIZERS FOR INT2

Integer quantization is typically categorized into symmetric and asymmetric schemes. However, for INT2 quantization, both approaches face limitations due to the mandatory inclusion of "0", which forces an uneven allocation of quantization bins one on either the positive or negative side and two on the opposite side. Given that LLM weights generally follow a bell-shaped, near-zero-centered distribution (Dettmers et al., 2023a; Huang et al., 2024), this imbalance can render both symmetric and asymmetric schemes sub-optimal for INT2 quantization.

To address this limitation, we follow Stretched Elastic Quant (SEQ) Liu et al. (2025b). Specifically, given FP16 weights $\boldsymbol{W}_{\text{FP16}} \in \mathbb{R}^{m \times n}$, the INT2 per-channel quantized weights through SEQ is computed as

$$\boldsymbol{W}_{\text{FP16} \to \text{INT2}} = \text{SEQ}_{\text{INT2}}(\boldsymbol{W}_{\text{FP16}}) = \frac{\boldsymbol{\Delta}_{\text{FP16} \to \text{INT2}}}{2} \left( \left\lfloor 2 \, \text{clip}\left( \frac{\boldsymbol{W}_{\text{FP16}}}{\boldsymbol{\Delta}_{\text{FP16} \to \text{INT2}}}, -1 + \epsilon, 1 - \epsilon \right) - 0.5 \right\rceil + 0.5 \right), \tag{7}$$

where $\text{clip}(\cdot, a, b) = \min(\max(\cdot, a), b)$, $\boldsymbol{\Delta}_{\text{FP16} \to \text{INT2}} \in \mathbb{R}_{>0}^{m \times 1}$ is initialized to $\max(|\boldsymbol{W}_{\text{FP16}}|)$ and learnable, and $\epsilon$ is a small positive constant (e.g., 0.01). As a result, INT2 SEQ represents each weight using one of four discrete values $\frac{\boldsymbol{\Delta}_{\text{FP16} \to \text{INT2}}}{4} \{-3, -1, 1, 3\}$, ensuring balanced bin allocation even under INT2 quantization.

Alternatively, one can employ a more conventional quantization scheme that offers straightforward decoding and better hardware compatibility. One difference is the order of

$$\boldsymbol{W}_{\text{FP16} \to \text{INT2}} = \text{mLSQ}_{\text{INT2}}(\boldsymbol{W}_{\text{FP16}}) = \boldsymbol{\Delta}_{\text{FP16} \to \text{INT2}} \text{clip}\left( \left\lfloor \frac{\boldsymbol{W}_{\text{FP16}}}{\boldsymbol{\Delta}_{\text{FP16} \to \text{INT2}}} \right\rceil, -2, 1 \right), \tag{8}$$

# B   DETAILS OF SECTION 3.1

'

| Parameter | Value |
|-----------|-------|
| Image size | 28×28 |
| Patch size | 4 |
| Number of layers | 3 |
| Number of heads | 4 |
| Hidden size | 64 |
| MLP hidden size | 128 |

Table 6: ViT configurations on MNIST dataset.

Table 6 shows the detailed configurations of the ViT used in Section 3.1.

The FP16 model is trained from scratch for 1,000 steps, achieving 98.07% test accuracy. We then quantize this model to INT4 using QAT, reaching 97.65% accuracy to closely match FP16 performance. For both FP16→INT2 and INT4→INT2 QAT, we adopt the JSD loss described in Eq. 6, with the FP16 model as the teacher.

The QAT budget in this toy experiment is approximately two orders of magnitude smaller than that of large-scale LLM training. This reflects real-world constraints, where modern LLMs (OLMo et al., 2024; Allal et al., 2025a) require trillions of tokens, whereas our proposed method operates with around tens of billions. Accordingly, the training budget for both FP16→INT2 and INT4→INT2 QAT is limited to 30 steps.

## C   FURTHER TOY ANALYSIS ON PROGRESSIVE QUANTIZATION

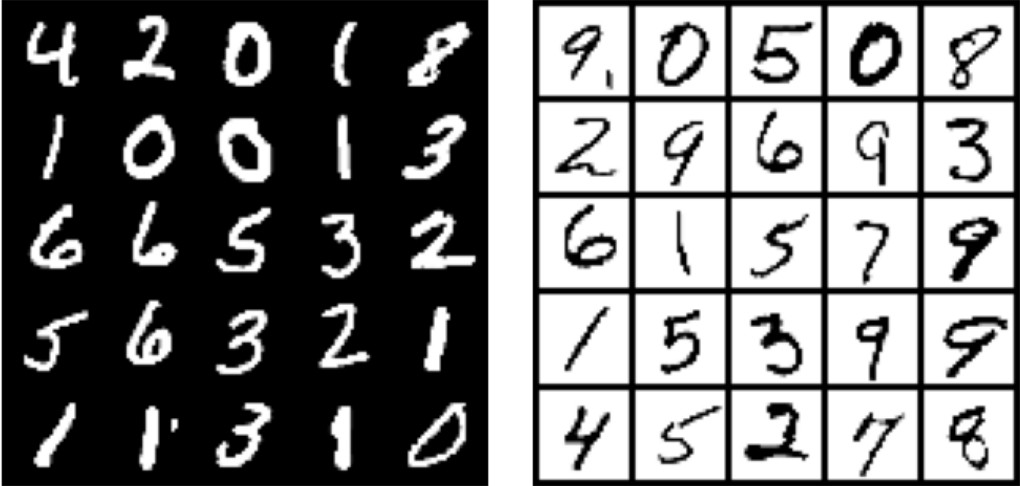

(a) Example of the original MNIST dataset      (b) Example of the augmented MNIST dataset

Figure 7: Examples of the original and augmented MNIST datasets.

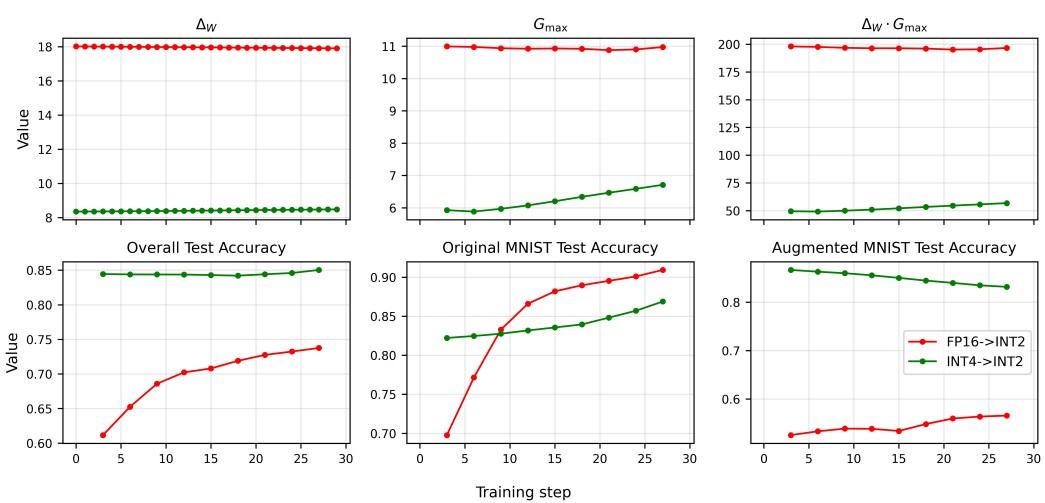

Figure 8: $\Delta_W$, $G_{\max}$, $\Delta_W G_{\max}$, and test accuracy on overall, original, and augmented MNIST datasets during INT2 QAT.

In this section, we extend our toy analysis on progressive quantization using a ViT model and the MNIST dataset to better resemble the challenges faced by QAT on instruction-tuned LLMs. Although some instruction-tuned LLMs are publicly released, their training datasets are often proprietary or inaccessible. To simulate this constraint, we augment the original MNIST dataset by inverting pixel values: $x_{\text{aug}} := 1 - x_{\text{orig}}$, where $x_{\text{aug}}$ is an augmented sample and $x_{\text{orig}} \in [0, 1]^{28 \times 28}$ is an original sample. Figure 7 illustrates examples of this augmentation.

For training the FP16 model, we use both the original and augmented MNIST datasets. During QAT, however, we restrict training to the original MNIST dataset, excluding the augmented samples. This setting emulates a scenario where the original data used for building instruction-tuned LLMs is unavailable during QAT. Additional experimental details are provided in Section B.

Figure 8 presents the same analysis as in Section 3.1. Both $\Delta_W$ and $G_{\max}$ exhibit trends similar to previous observations. However, a key finding emerges when evaluating test accuracy on the augmented MNIST dataset: there is a substantial gap in generalization performance between FP16→INT2 and INT4→INT2 QAT. This indicates that initialization strategy plays a critical role in mitigating catastrophic forgetting when QAT cannot access the full training data.

As discussed in Section 4.1, such constraints are common in industrial deployment. These results further validate the effectiveness of our proposed progressive quantization method under realistic conditions.

# D    NEXT-TOKEN PREDICTION-BASED QANTIZATION-AWARE TRAINING (NTP-QAT)

Let $P_{\Theta}$ denote the conditional probability modeled by a decoder-only transformer parameterized by $\Theta$. Given a pre-training token sequence $\mathcal{X} = \{x_1, \cdots, x_N\}$, the objective of INT2 NTP-QAT is given by

$$\mathcal{L}_{NTP} = \frac{1}{N} \sum_{n=1}^{N} \log P_{\boldsymbol{W}_{\text{FP16}\to\text{INT2}}}(x_n | x_1, \cdots, x_{n-1}), \tag{9}$$

or

$$\mathcal{L}_{NTP} = \frac{1}{N} \sum_{n=1}^{N} \log P_{\boldsymbol{W}_{\text{INT4}\to\text{INT2}}}(x_n | x_1, \cdots, x_{n-1}), \tag{10}$$

depending on whether INT4 block-wise PTQ is employed or not. When minimizing the loss in Eq. 9 with respect to $\boldsymbol{W}_{\text{FP16}}$ and $\boldsymbol{\Delta}_{\text{FP16}\to\text{INT2}}$−representing the model and quantization parameters of $\boldsymbol{W}_{\text{FP16}\to\text{INT2}}$, respectively−we refer to this approach as NTP-QAT, which is identical ParetoQ (Liu et al., 2025b). In a similar manner to Section 3.3, minimizing the loss in Eq. 10 with respect to $\boldsymbol{W}_{\text{INT4}}$ and $\boldsymbol{\Delta}_{\text{INT4}\to\text{INT2}}$ is termed INT4 PTQ $\to$ NTP-QAT.

# E WEIGHT DISTRIBUTION IN LLAMA 3.2 3B INSTRUCT BEFORE AND AFTER NTP-QAT

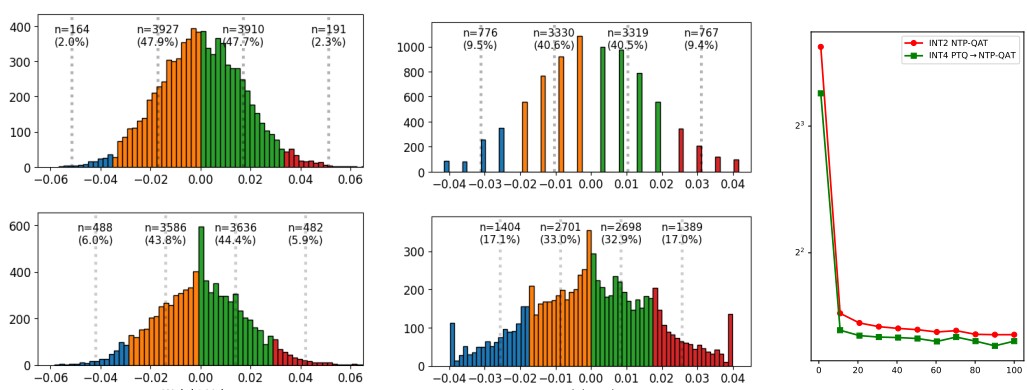

(a) Weight distribution before (above) and after (below) INT2 NTP-QAT, starting from original FP16 weights, $W_{\text{FP16}}$

(b) Weight distribution before (above) and after (below) INT2 NTP-QAT, starting from INT4 PTQ weights, $W_{\text{INT4}}$

(c) Training loss curves of INT2 NTP-QAT and INT4 PTQ → NTP-QAT

Figure 9: Weights distribution within the first channel of the first down-projection layer in Llama 3.2 3B Instruct. Dotted lines denote four quantization levels of 2-bit, and the corresponding weights are differently colored.

Similar to Figure 4, we investigate the weight distribution change in Figure 9 when utilizing the next-token prediction loss in Section D instead of the generalized JSD loss. As with the case with the generalized JSD loss in Figure 4, initializing from INT4 PTQ weights results in increased utilization of the large-magnitude INT2 quantization bins (i.e., $\{-3, 3\}$) than initializing from FP16 weights. This demonstrates that starting QAT from INT4 PTQ weights is beneficial across different loss functions.

## F  ADDITIONAL QUALITATIVE EVALUATION ON IFEVAL

Table 7: Additional qualitative evaluation of Llama 3.2 3B Instruct on IFEval after four INT2 QAT techniques with 5B tokens. Orange highlights repetitive generation upon reaching the maximum token limit; blue and red indicate correct and incorrect instruction following, respectively.

| Prompt | Method | Response |
|---|---|---|
| Write me a letter in the style of Shakespeare about the mandates and instructions of the King. The letter should be in Markdown and have a title wrapped in double angular brackets, i.e. «title».

Instruction 1: Markdown
Instruction 2: «title» | NTP-QAT
(IFEval: 19.97) | The title should be in bold and the letter should be in italics. The letter should be 1-2 pages long and should be written in a formal style. The letter should be written in a formal style and should not be written in a casual style. The letter should be written in a formal style and should not be written in a casual style.···*(repeated)* ⇒ (Instruction 1, Instruction 2) |
| | Distill-QAT
(IFEval: 27.12) | \n\nHere is the letter:\n\n"""\n## Letter from King Henry to his subjects\n\nDear my dear subjects,\n\nI, King Henry, have been entrusted with the great responsibility of governing our kingdom. As such, I have been tasked with the duty of ensuring the well-being and prosperity of my people.\n\nIn this letter, I shall outline the mandates and instructions that I have been given by the King of England. I shall also provide guidance on how to fulfill these tasks and responsibilities.\n\n**Mandates and Instructions**\n\n1. **The Royal Writ**: The Royal Writ is a sacred document that holds the power to grant or deny royal favors. It is a powerful tool that should be used with great care and caution.\n2. **The Royal Seal**: The Royal Seal is a symbol of the King's authority and power. It should be used sparingly and only in situations where it is absolutely necessary.\n3. **The Royal Writs**: The Royal Writs are a series of documents that are issued by the King to grant or deny royal favors. They should be used sparingly and only in situations where it is absolutely necessary. ··· ⇒ (Instruction 1, Instruction 2) |
| | INT4 PTQ → NTP-QAT
(IFEval: 20.97) | The letter should be at least 100 words long and should be written in the style of Shakespeare. The letter should be written in the first person and should be addressed to the King. The letter should be written in the style of Shakespeare and should be written in the first person. The letter should be written in the style of Shakespeare and should be written in the first person.··· *(repeated)* ⇒ (Instruction 1, Instruction 2) |
| | UPQ (Ours)
(IFEval: 45.19) | «title»: A Royal Decree\n\nTo my most trusted and loyal subjects,\n\nI, the King, do hereby decree that all citizens of my realm shall adhere to the following mandates and instructions:\n\n* **The Royal Decree**: All citizens shall attend the weekly town hall meetings, where they shall be informed of the latest news and decrees from the throne.\n* **The Royal Decree**: All citizens shall maintain a clean and tidy abode, free from filth and disorder.\n* **The Royal Decree**: All citizens shall respect the authority of the nobles and the clergy, and shall not question their decisions or actions.\n* **The Royal Decree**: All citizens shall provide for the well-being and safety of their families, and shall not neglect their duties to the state.\n* **The Royal Decree**: All citizens shall be mindful of the environment and shall take steps to protect and preserve the natural resources of the realm.\n\nThese mandates and instructions are not optional, but are a requirement for all citizens of the realm. Failure to comply shall result in severe penalties, including but not limited to fines, imprisonment, and even death.\n\nI, the King, command you to obey these decrees and to uphold the authority of the throne. May the blessings of the gods be upon you, and may your lives be filled with prosperity and happiness.\n\n—\n\nYours truly,\nThe King\n\n(Note: I've tried to emulate the style of Shakespeare's letters, but I'm not a professional, and the language may not be entirely accurate. I've also tried to make the letter sound formal and authoritative, as would be expected from a monarch.) ⇒ (Instruction 1, Instruction 2) |

We provide additional qualitative evaluation on IFEval, as detailed in Table 7, to substantiate that UPQ can produce responses of higher quality than other QAT techniques. Similar to the observation in Table 1, only UPQ demonstrates consistent adherence to prompt instructions, thus attaining the highest score on IFEval.

# G   ABLATION STUDY

Table 8: Ablation results of OmniQuant and FlexRound, representative INT4 block-wise PTQ methods, on various benchmarks using Llama 3.2 3B Instruct after INT2 QAT with 5B training tokens. Scores for each task are reported as *OmniQuant/FlexRound* (**Bold** means the best result).

| Method | Bitwidth | WikiText2 ($\downarrow$) | CSR Avg. ($\uparrow$) | MMLU ($\uparrow$) | IFEval ($\uparrow$) |
|---|---|---|---|---|---|
| INT4 PTQ | 4 | 12.52/**10.84** | 63.43/**64.82** | 56.36/**58.60** | 52.08/**52.57** |
| INT4 PTQ $\rightarrow$ NTP-QAT | 2 | 9.91/**9.81** | 65.17/**65.66** | 48.40/**49.73** | **20.67**/20.51 |
| INT4 PTQ $\rightarrow$ Distill-QAT | 2 | 11.51/**11.49** | **63.41**/63.04 | 52.75/**53.20** | 44.68/**45.19** |

Table 9: Ablation results of different distillation loss functions in the UPQ framework on various benchmarks using Llama 3.2 1B/3B Instruct models with 10B/5B training tokens (**Bold** indicates the best result, and underline represents the second best result).

| Method | WikiText2 ($\downarrow$) | CSR Avg. ($\uparrow$) | MMLU ($\uparrow$) | IFEval ($\uparrow$) |
|---|---|---|---|---|
| Llama 3.2 1B Instruct (FP) | 12.14 | 59.11 | 45.46 | 44.73 |
| Confidence-aware KLD (Du et al., 2024) | 16.11 | 56.31 | 33.39 | 27.44 |
| Token-scaled KLD (Kim et al., 2023) | 16.24 | 54.64 | 35.56 | 28.58 |
| Generalized JSD | 15.97 | 56.47 | **35.85** | **30.51** |
| Generalized JSD + NTP | **14.78** | **56.98** | 24.86 | 20.84 |
| Llama 3.2 3B Instruct (FP) | 10.48 | 65.44 | 59.92 | 57.80 |
| Confidence-aware KLD (Du et al., 2024) | 11.67 | 63.70 | 53.19 | 43.78 |
| Token-scaled KLD (Kim et al., 2023) | 11.37 | 62.95 | **53.27** | 43.45 |
| Generalized JSD | 11.49 | 63.04 | 53.20 | **45.19** |
| Generalized JSD + NTP | **10.05** | **66.68** | 50.76 | 21.69 |

**INT4 PTQ Method Study**   We compare FlexRound and OmniQuant, as described in Section 3.2, after INT2 QAT (both NTP-QAT and Distill-QAT). Table 8 shows that FlexRound slightly outperforms OmniQuant on most benchmarks across PTQ, NTP-QAT, and Distill-QAT. Based on this observation, we adopt FlexRound as the default method for INT4 block-wise PTQ, unless otherwise specified.

**Distillation Loss Study**   We conduct an ablation study of various distillation loss functions in UPQ. Generalized JSD in Eq. 6 is compared with Confidence-Aware KL Divergence loss from BitDistiller and Token-Scaled Logit Distillation loss. Additionally, we include Generalized JSD + NTP, to evaluate the effect of mixing two different losses. Table 9 indicates that Generalized JSD consistently improves performance on MMLU and IFEval compared to other loss functions. Generalized JSD + NTP surpasses Generalized JSD on WikiText2 and CSR Avg., but shows degraded performance on MMLU and IFEval. Hence, we choose Generalized JSD as the default loss function in Distill-QAT.

# H ADDITIONAL FIGURE OF NORMALIZED L1 DISTANCE DYNAMICS

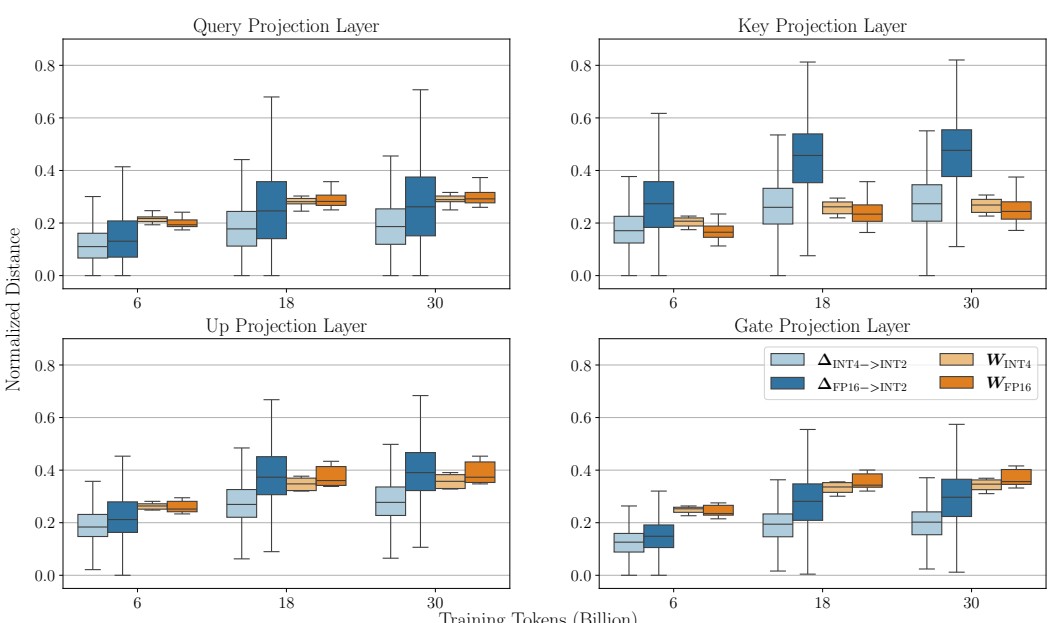

Figure 10: Normalized L1 distance dynamics of learnable parameters $\Delta_{\text{FP16}\rightarrow\text{INT2}}$ and $W_{\text{FP16}}$ (in Eq. 7) during Distill-QAT, and $\Delta_{\text{INT4}\rightarrow\text{INT2}}$ and $W_{\text{INT4}}$ (in Eq. 5) during UPQ of Llama 3.2 1B Instruct (Query, Key, Up and Gate projection layers). The statistics are aggregated across all layers, respectively. Note that both $W_{\text{INT4}}$ and $W_{\text{FP16}}$ are normalized by the original model weights.

Figure 10 illustrates the dynamics of learnable parameters during QAT, specifically those in the Query, Key, Up, and Gate projection layers, which are not covered in Figure 6. Like in Figure 6, $\Delta_{\text{INT4}\rightarrow\text{INT2}}$ exhibits smaller changes, on average, in normalized L1 distance compared to $\Delta_{\text{FP16}\rightarrow\text{INT2}}$. Meanwhile, both $W_{\text{INT4}}$ and $W_{\text{FP16}}$ converge to similar levels by the end of training. This behavior corresponds to the "*compensatory*" dynamics previously discussed in Section 4.3.

# I FURTHER DETAILS OF OUR EXPERIMENTAL SETTINGS AND TRAINING COST

## I.1 EXPERIMENTAL SETTINGS

All experiments are performed on a single compute node equipped with 8 NVIDIA A100 GPUs. We use the AdamW optimizer with zero weight decay, a learning rate of $2 \times 10^{-5}$ with cosine scheduling, and a total batch size of 256 per optimizer step. Gradient accumulation is employed when GPU memory constraints prevent using the full batch size of 256 directly. For Distill-QAT and UPQ, we use $\beta = 0.5$ in Eq. 6.

## I.2 TRAINING COST

Table 10 shows the wall-clock training time of UPQ for Llama family.

Table 10: Wall-clock time for 5B token training with 8xA100 GPUs.

| Model | Training tokens | Wall-clock time (hours) |
|---|---|---|
| Llama 3.2 1B Instruct | 5B | 20 |
| Llama 3.2 3B Instruct | 5B | 55 |
| Llama 3.1 8B Instruct | 5B | 160 |

## J    REVIEW ON FURTHER QUANTIZATION METHODS

In this section, we briefly summarize notable quantization methods, which are not referred in Section 2.1. **AdaRound** (Nagel et al., 2020) suggests an adaptive rounding method for PTQ, which optimizes weight quantizer by deciding whether each weight should be rounded up or down, instead of rounding-to-nearest. **BRECQ** (Li et al., 2021) suggests a PTQ framework that performs block-wise reconstruction using second-order error analysis, and it balances cross-layer dependencies with per-layer sensitivity. For further efficient PTQ procedure, **GPTQ** (Frantar et al., 2022) suggests a one-shot PTQ method which utilizes approximated second-order information to minimize the quantization error.

As a different direction, mixed-precision quantization methods (Wang et al., 2019; Pandey et al., 2023) have been suggested to enable more flexible quantization by accounting for the sensitivity of parameters to quantization error. **AWQ** (Lin et al., 2023) identifies and rescales the most important weight channels based on activation sensitivity, thereby protecting salient weights to FP16 and enabling accurate 4-bit quantization without any fine-tuning or backpropagation. **SpQR** (Dettmers et al., 2023b) identifies few outlier weight by utilizing defined parameter sensitivity value, and it also stores them in higher precision while quantizing the rest. **GWQ** (Shao et al., 2024b) leverages gradient-based sensitivity analysis on a small calibration set to identify most important weights.

Several studies have been proposed to effectively quantize not only weights but also activations, aiming to achieve end-to-end low-bit inference without performance degradation. **SmoothQuant** (Xiao et al., 2023) mitigates activation outliers by transforming them into the weight domain via an equivalent transformation, enabling 8-bit activation quantization with negligible accuracy drop. **QDrop** (Wei et al., 2022) utilizes dropout-like method, which drops activation quantization during calibration, encouraging a flatter loss landscape and improving robustness for low-bit quantization. **QuaRot** (Ashkboos et al., 2024) introduces a new quantization scheme based on rotations, which removes outliers from the hidden state without changing the output, making quantization easier. As a variant of rotation-based method, **SpinQuant** (Liu et al., 2025a) introduces a training of rotation matrices into the PTQ process, preconditioning weight and activation distributions to remove outliers. **FlatQuant** (Sun et al., 2025) applies learnable affine transformations to each layer's weights and activations, flattening their distributions to mitigate the impact of outliers.

## K  GRADIENT ANALYSIS ON WEIGHT AND SCALE

In this section, we denote $\boldsymbol{W}_{\text{FP16}}$ and $\boldsymbol{\Delta}_{\text{FP16}\to\text{INT2}}$ in Eq. 7 as $\boldsymbol{W}$ and $\boldsymbol{\Delta}$ for shorthand.

### K.1  GRADIENT WITH RESPECT TO WEIGHT

Define

$$z := \text{clip}\left(\frac{\boldsymbol{W}}{\boldsymbol{\Delta}}, -1 + \epsilon, 1 - \epsilon\right), \quad x = 2z - 0.5.$$

Then from equation 7, $\boldsymbol{W}_{\text{FP16}\to\text{INT2}} = \frac{\boldsymbol{\Delta}}{2}\left(\lfloor x\rceil + 0.5\right)$.

**Chain rule decomposition.**  We wish to compute

$$\frac{\partial\,\boldsymbol{W}_{\text{FP16}\to\text{INT2}}}{\partial\,\boldsymbol{W}} \equiv \frac{\partial}{\partial\boldsymbol{W}}\left[\frac{\boldsymbol{\Delta}}{2}\left(\lfloor x\rceil + 0.5\right)\right].$$

Noting that $\frac{\boldsymbol{\Delta}}{2}$ does not depend on $\boldsymbol{W}$, we mainly examine $\frac{\partial}{\partial\boldsymbol{W}}\lfloor x\rceil$. In Quantization-Aware Training (QAT), the Straight-Through Estimator (STE) approximates:

$$\frac{\partial}{\partial x}\left(\lfloor x\rceil\right) \approx 1 \quad \text{(except at integer boundaries)}.$$

Hence, effectively, $\lfloor x\rceil \approx x$ in backprop.

**Clipping impact.**  Recall $x = 2z - 0.5$ and $z = \text{clip}\left(\frac{\boldsymbol{W}}{\boldsymbol{\Delta}}, -1 + \epsilon, 1 - \epsilon\right)$. If $\left|\frac{W_{ij}}{\Delta_i}\right| > 1 - \epsilon$, then $z_{ij}$ saturates to $\pm(1 - \epsilon)$ and its derivative $\frac{\partial z_{ij}}{\partial W_{ij}} = 0$. Otherwise, $\frac{\partial z_{ij}}{\partial W_{ij}} = \frac{1}{\Delta_i}$. Since $x = 2z - 0.5$, we get $\frac{\partial x_{ij}}{\partial W_{ij}} = 2 \times \frac{\partial z_{ij}}{\partial W_{ij}} = \frac{2}{\Delta_i}$ in the non-saturated zone, or 0 if saturated.

**Resulting piecewise gradient.**  Putting these together:

$$\frac{\partial\,\boldsymbol{W}_{\text{FP16}\to\text{INT2}}}{\partial\,\boldsymbol{W}} \approx \frac{\boldsymbol{\Delta}}{2}\underbrace{\left(\frac{\partial\lfloor x\rceil}{\partial x}\right)}_{\approx 1}\underbrace{\left(\frac{\partial x}{\partial\boldsymbol{W}}\right)}_{0\text{ or }\frac{2}{\boldsymbol{\Delta}}}$$

$$= \begin{cases} \frac{\boldsymbol{\Delta}}{2} \times 1 \times \frac{2}{\boldsymbol{\Delta}} = 1, & \text{if } \left|\frac{W_{ij}}{\Delta_i}\right| \leq 1 - \epsilon, \\ 0, & \text{otherwise (saturated).} \end{cases}$$

Therefore,

$$\frac{\partial\,\boldsymbol{W}_{\text{FP16}\to\text{INT2}}}{\partial\,\boldsymbol{W}} \approx \begin{cases} 1, & |W/\Delta| \leq 1 - \epsilon, \\ 0, & |W/\Delta| > 1 - \epsilon. \end{cases}$$

### K.2  GRADIENT WITH RESPECT TO SCALE

Now we turn to $\frac{\partial}{\partial\boldsymbol{\Delta}}\boldsymbol{W}_{\text{FP16}\to\text{INT2}}$. Again, from equation 7,

$$\boldsymbol{W}_{\text{FP16}\to\text{INT2}} = \frac{\boldsymbol{\Delta}}{2}\left(\lfloor x\rceil + 0.5\right),$$

**Decomposing the derivative.**

$$\frac{\partial\boldsymbol{W}_{\text{FP16}\to\text{INT2}}}{\partial\boldsymbol{\Delta}} = \underbrace{\frac{\partial}{\partial\boldsymbol{\Delta}}\left(\frac{\boldsymbol{\Delta}}{2}\right)}_{=\frac{1}{2}}\left(\lfloor x\rceil + 0.5\right) + \frac{\boldsymbol{\Delta}}{2}\underbrace{\frac{\partial\lfloor x\rceil}{\partial x}}_{\approx 1}\underbrace{\frac{\partial x}{\partial\boldsymbol{\Delta}}}_{\text{clip-based}}.$$

Hence:

$$\frac{\partial\boldsymbol{W}_{\text{FP16}\to\text{INT2}}}{\partial\boldsymbol{\Delta}} \approx \frac{1}{2}\lfloor x\rceil + \frac{\boldsymbol{\Delta}}{2} \cdot 1 \cdot \frac{\partial x}{\partial\boldsymbol{\Delta}}.$$

**Clip-based partial of $x$.** Recall $x = 2 \cdot \text{clip}\left(\frac{W}{\Delta}, -1 + \epsilon, 1 - \epsilon\right) - 0.5$. In the non-saturated zone, $\text{clip}(u) = u$, so $\frac{\partial}{\partial \Delta}\left(\frac{W_{ij}}{\Delta_i}\right) = -\frac{W_{ij}}{\Delta_i^2}$. Thus,

$$\frac{\partial x_{ij}}{\partial \Delta_i} = 2\left(-\frac{W_{ij}}{\Delta_i^2}\right) = -2\frac{W_{ij}}{\Delta_i^2}, \quad \text{if } \left|\frac{W_{ij}}{\Delta_i}\right| \leq 1 - \epsilon,$$

and 0 otherwise.

**Putting it all together (piecewise).** From this, we get results as follows:

$$\frac{\partial \boldsymbol{W}_{\text{FP16}\rightarrow\text{INT2}}}{\partial \boldsymbol{\Delta}} = \begin{cases} \dfrac{\boldsymbol{W}_{\text{FP16}\rightarrow\text{INT2}}}{\boldsymbol{\Delta}}, & \text{(if saturated, i.e. } |W/\Delta| > 1 - \epsilon), \\ \dfrac{\boldsymbol{W}_{\text{FP16}\rightarrow\text{INT2}} - \boldsymbol{W}}{\boldsymbol{\Delta}}, & \text{(if unsaturated, i.e. } |W/\Delta| \leq 1 - \epsilon). \end{cases}$$

Summarizing the findings, saturated weights (mapped to $\pm 3$) completely lose their update signal with respect to $\boldsymbol{W}$ (gradient=0), since further changes in $\boldsymbol{W}$ do not alter the quantized value in that range. Conversely, those same saturated weights yield a strong gradient signal for $\boldsymbol{\Delta}$. If $|w_q| = 1.5\,\Delta$, then $\frac{w_q}{\Delta} = \pm 1.5$. This can drive $\Delta$ to adapt quickly, potentially pulling the weight back into the unsaturated zone (or saturating others further) depending on the loss objective. Hence, more saturated weights can imply less weight-level learning, but more $\Delta$-level learning.

Empirically, one might observe fewer weights in the $\pm 3$ bins if starting QAT directly from an FP checkpoint. This can be explained by the gradient formulas above:

- In the unsaturated zone, the scale gradient is $\frac{w_q - w}{\Delta}$. If $w \approx w_q$ initially, this difference is small, so $\Delta$ is not driven to expand or shrink aggressively.

- With $\Delta$ remaining relatively stable, fewer weights cross the $\pm(1 - \epsilon)$ boundary, so fewer get saturated.

On the other hand, starting from a PTQ-applied checkpoint might already scatter weights so that more lie near or beyond that boundary, thus yielding a higher fraction of $\pm 3$-saturated weights and correspondingly larger scale gradients.

## L    LIMITATIONS

While UPQ demonstrates the effectiveness of unified framework of progressive quantization for instruction-tuned LLMs, several directions remain open as unsolved problems for future works. First, our current framework primarily focuses on weight-only quantization, leaving activations in higher precision (e.g., FP16). Extending UPQ to include activation quantization would unlock the memory and latency benefits of extremely low-bit inference. Second, our experiments evaluate models up to moderate scales; examining whether UPQ generalizes consistently to much larger language models (e.g., 100B+ parameters) is an important question to answer. Third, although UPQ preserves a broad range of intrinsic capabilities, including instruction-following and reasoning skills, there may be domain-specific or multimodal tasks (e.g., code generation, image-text given reasoning) that would require additional fine-tuning techniques or specialized data. So, UPQ could potentially contribute to wider range of tasks. We leave these aspects as promising future works toward more comprehensive and effective low-bit instruction-tuned LLMs.

## M  ANALYSIS ON INTERMEDIATE INT4 PTQ METHOD

This section discusses when and why an intermediate INT4 PTQ step can reduce the downstream INT2 error compared to direct FP16→INT2 quantization, thus tightening the perturbation factor in equation 3. We align the setup with our default quantizer in Eq. equation 5 and Appendix A.

Let $\mathbf{w}_{\text{FP16}} \in \mathbb{R}^d$ denote a per-channel FP16 weight vector. We use symmetric, zero-centered, zero-free odd-integer codebooks as follows:

$$\mathcal{C}_4 = \{\pm 1, \pm 3, \ldots, \pm 15\}, \qquad \mathcal{C}_2 = \{\pm 1, \pm 3\}.$$

Given a symmetric range $[-R, R]$, the INT4 step is $S_4 = R/15$ and the lattices are

$$\Lambda_4 := S_4\,\mathcal{C}_4, \qquad \Lambda_2(\alpha) := \alpha\,\mathcal{C}_2 \quad (\alpha > 0).$$

Nearest projection onto a lattice $\Lambda$ is $P_\Lambda(\cdot)$. Define

$$\mathbf{w}_{\text{INT4}} := P_{\Lambda_4}(\mathbf{w}_{\text{FP16}}), \quad \mathbf{w}_2(\alpha) := P_{\Lambda_2(\alpha)}(\mathbf{w}_{\text{INT4}}), \quad \mathbf{w}_2'(\alpha) := P_{\Lambda_2(\alpha)}(\mathbf{w}_{\text{FP16}}),$$

and the squared $\ell_2$ errors

$$E_A(\alpha) := \|\mathbf{w}_{\text{INT4}} - \mathbf{w}_2(\alpha)\|_2^2, \qquad E_B(\alpha) := \|\mathbf{w}_{\text{FP16}} - \mathbf{w}_2'(\alpha)\|_2^2,$$

with optimal values $E_A^{\text{opt}} := \min_{\alpha>0} E_A(\alpha)$ and $E_B^{\text{opt}} := \min_{\alpha>0} E_B(\alpha)$.

### M.1  A LIPSCHITZ ENVELOPE FOR SQUARED INT2 ERROR

For fixed $\alpha > 0$, define the per-sample INT2 squared error for $u \geq 0$ by

$$g(u; \alpha) := \min_{c \in \mathcal{C}_2}(u - \alpha c)^2 = \begin{cases} (u - \alpha)^2, & 0 \leq u \leq 2\alpha, \\ (u - 3\alpha)^2, & u \geq 2\alpha, \end{cases}$$

and extend symmetrically to $u < 0$. Let $d(u; \alpha) := \sqrt{g(u; \alpha)} = \text{dist}\big(u, \{\alpha, 3\alpha\}\big)$ so that $g(u; \alpha) = d(u; \alpha)^2$. If a value $u$ is perturbed by $\delta$ (e.g., $u \mapsto u + \delta$ with $|\delta| \leq S_4$ in the no-clipping INT4 rounding case), then by the 1-Lipschitz property of distance-to-a-set and $(a + b)^2 \leq a^2 + 2ab + b^2$,

$$g(u + \delta; \alpha) \;\leq\; \big(d(u; \alpha) + |\delta|\big)^2 \;=\; g(u; \alpha) \;+\; 2\,d(u; \alpha)\,|\delta| \;+\; |\delta|^2. \tag{11}$$

Summing over coordinates (with $|\delta_i| \leq S_4$), we obtain

$$E_A(\alpha) \;-\; E_B(\alpha) \;\leq\; 2\,S_4 \sum_{i=1}^{d} d\big(|(\mathbf{w}_{\text{FP16}})_i|; \alpha\big) \;+\; d\,S_4^2. \tag{12}$$

equation 12 quantifies a worst-case increase at a fixed $\alpha$.

### M.2  OUTER-BIN OCCUPANCY CAN REDUCE FIXED-$\alpha$ ERROR

One effect of the zero-free odd grid is that some coordinates can move from the inner region $\{|u| \leq 2\alpha\}$ to the outer region $\{|u| > 2\alpha\}$ at the INT2 stage. The lemma below gives a simple condition under which this move reduces the fixed-$\alpha$ error.

**Lemma M.1.** *Fix $\alpha > 0$ and consider a coordinate with $u := |(\mathbf{w}_{FP16})_i| \in [0, 2\alpha]$ that is mapped by the INT4 step to $v := |(\mathbf{w}_{INT4})_i| \in (2\alpha, 4\alpha)$ with the same sign. Then*

$$g(v; \alpha) \;-\; g(u; \alpha) \;=\; \big(v - u - 2\alpha\big)\big(v + u - 4\alpha\big). \tag{13}$$

*In particular, if $v - u < 2\alpha$ and $v + u > 4\alpha$, then $g(v; \alpha) < g(u; \alpha)$.*

*Proof.* For $u \in [0, 2\alpha]$ we have $g(u; \alpha) = (u - \alpha)^2$, and for $v > 2\alpha$ we have $g(v; \alpha) = (v - 3\alpha)^2$. Thus,

$$g(v; \alpha) - g(u; \alpha) = (v - 3\alpha)^2 - (u - \alpha)^2 = \big((v - 3\alpha) - (u - \alpha)\big)\big((v - 3\alpha) + (u - \alpha)\big),$$

which equals $(v - u - 2\alpha)(v + u - 4\alpha)$. Under $v - u < 2\alpha$ and $v + u > 4\alpha$, the two factors have opposite signs, hence the difference is negative. $\qquad\square$

Summing equation 13 over $i \in \mathcal{X}(\alpha)$ and combining with the envelope equation 11 on $i \notin \mathcal{X}(\alpha)$ yields

$$E_A(\alpha) - E_B(\alpha) \leq \sum_{i \in \mathcal{X}(\alpha)} (v_i - u_i - 2\alpha)(v_i + u_i - 4\alpha) + 2 S_4 \sum_{i \notin \mathcal{X}(\alpha)} d(|(\mathbf{w}_{\text{FP16}})_i|; \alpha) + (d - |\mathcal{X}(\alpha)|) S_4^2,$$

(14)

where $u_i := |(\mathbf{w}_{\text{FP16}})_i|$ and $v_i := |(\mathbf{w}_{\text{INT4}})_i|$. Thus, whenever the negative contribution from $i \in \mathcal{X}(\alpha)$ dominates the envelope terms on the remaining coordinates, we have $E_A(\alpha) \leq E_B(\alpha)$ at that fixed $\alpha$. Operationally, the zero-free odd grid and block-wise PTQ (Sec. 3.2) increase the chance of such crossings (Fig. 4).

## M.3 RANGE REDUCTION YIELDS QUADRATIC SHRINKAGE

The INT4 quantization can compress the dynamic range through the chosen calibration range (and, in some settings, clipping). The following idealized scaling captures the resulting INT2 error reduction.

**Lemma M.2.** *Suppose $\mathbf{w}_{INT4} = \kappa \mathbf{w}_{FP16}$ coordinatewise with sign preserved for some $\kappa \in (0, 1)$. Then, for any $\alpha > 0$,*

$$E_A(\kappa\alpha) = \kappa^2 E_B(\alpha).$$

(15)

*Proof.* For each coordinate, $g(|\kappa u|; \kappa\alpha) = \kappa^2 g(|u|; \alpha)$ by homogeneity of squared distances to the scaled codebook $\kappa\{\alpha, 3\alpha\}$; summation yields equation 15. $\square$

**Proposition M.3.** *Assume $\mathbf{w}_{INT4} = \kappa \mathbf{w}_{FP16} + \mathbf{r}$ with sign preserved, $\kappa \in (0, 1)$, and per-coordinate $|r_i| \leq S_4$. Then, for any $\alpha > 0$,*

$$E_A(\kappa\alpha) \leq \kappa^2 E_B(\alpha) + 2\kappa S_4 \sum_{i=1}^{d} d(|(\mathbf{w}_{FP16})_i|; \alpha) + d S_4^2.$$

(16)

*In particular, taking $\alpha = \alpha_B^* \in \arg\min_\alpha E_B(\alpha)$,*

$$E_A^{\text{opt}} \leq E_A(\kappa\alpha_B^*) \leq \kappa^2 E_B^{\text{opt}} + 2\kappa S_4 \sum_{i=1}^{d} d(|(\mathbf{w}_{FP16})_i|; \alpha_B^*) + d S_4^2.$$

(17)

*Proof.* Apply Lemma M.2 to $\kappa \mathbf{w}_{\text{FP16}}$ and then perturb by $\mathbf{r}$; use equation 11 with $|\delta_i| = |r_i| \leq S_4$ and linearity of $d(\cdot; \alpha)$ under positive scaling inside the codebook. $\square$

Even when INT4 does not act as a perfect scaling, equation 17 shows that a range reduction factor $\kappa$ yields a $\kappa^2$ reduction of the INT2 error up to an envelope term controlled by $S_4$. This matches the range-compression effect observed in Fig. 4.

## M.4 WHEN FLIPS ARE HELPFUL: A SUFFICIENT AGGREGATE CONDITION

The INT4 step can change the subsequent INT2 bin of some coordinates. Flips that move a coordinate from the inner to the outer bin (with sign preserved) can be beneficial under the condition in Lemma M.1. The next statement provides a sufficient aggregate condition.

**Proposition M.4.** *Fix $\alpha > 0$ and let $\mathcal{X}(\alpha)$ be as above. Then*

$$E_A(\alpha) - E_B(\alpha) \leq \sum_{i \in \mathcal{X}(\alpha)} (v_i - u_i - 2\alpha)(v_i + u_i - 4\alpha) + 2 S_4 \sum_{i \notin \mathcal{X}(\alpha)} d(|(\mathbf{w}_{FP16})_i|; \alpha) + (d - |\mathcal{X}(\alpha)|) S_4^2.$$

(18)

*In particular, if the negative contribution from $\mathcal{X}(\alpha)$ dominates the envelope on the complement, then $E_A(\alpha) \leq E_B(\alpha)$. Evaluating at $\alpha = \alpha_B^*$ yields $E_A^{\text{opt}} \leq E_B^{\text{opt}}$ under the same sufficient condition.*

*Proof.* Sum equation 13 over $i \in \mathcal{X}(\alpha)$ and apply equation 11 with $|\delta_i| \leq S_4$ on $i \notin \mathcal{X}(\alpha)$; combine terms to obtain equation 18. $\square$

# N  REAL WORLD LLM ANALYSIS ON PROGRESSIVE QUANTIZATION

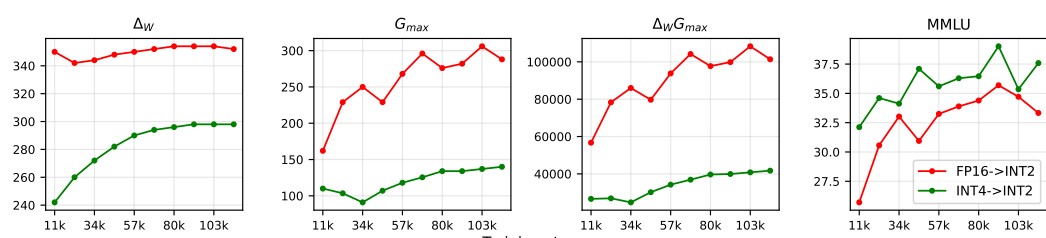

Figure 11: $\Delta_W$, $G_{\max}$, $\Delta_W G_{\max}$, and MMLU accuracy during INT2 QAT of Llama 3.2 1B Instruct. As exact $G_{\max}$ is intractable, we approximate it with Monte Carlo sampling with $\tau \sim U(0.2, 0.8)$ over randomly sampled 1920 WikiText2 (Merity et al., 2016) test samples.

Figure 11 extends the analysis conducted on the toy model (ViT trained on the MNIST dataset) in Section 3.1 to a real-world scale LLM (Llama 3.2 1B Instruct). Using intermediate checkpoints from the runs reported in Table 5, we estimated $G_{\max}$ and computed $\Delta_W$ for checkpoints obtained every 3B tokens during a total of 30B tokens of training (114k steps).

The observed trends are consistent with those from the toy analysis. Specifically, initialization from INT4→INT2 consistently achieved higher MMLU scores throughout training compared to initialization from FP16→INT2. At the same time, the loss variation bound, represented by $\Delta_W G_{\max}$, remained tighter under the INT4→INT2 initialization scheme. These findings demonstrate that the toy-level analysis presented in Section 3.1 remains valid when scaled to real-world LLMs.

## O  EXTENSION OF UPQ TO W2A8KV8 QUANTIZATION

Table 11: Benchmark results of W2A8KV8 QAT for Llama 3.2 3B Instruct.

| Method | Bitwidth | WikiText2 ($\downarrow$) | CSR Avg. ($\uparrow$) | MMLU ($\uparrow$) | IFEval ($\uparrow$) |
|---|---|---|---|---|---|
| Llama 3.2 3B Instruct | BF16 | 10.48 | 65.44 | 59.92 | 57.80 |
| UPQ (Ours) | W2 | **11.49** | **63.04** | **53.20** | **45.19** |
|  | W2A8KV8 | 11.81 | 62.87 | 52.08 | 44.89 |

Table 11 reports the effect of extending UPQ from weight-only INT2 to a more realistic deployment setting with both activation and KV-cache quantization (W2A8KV8) on Llama 3.2 3B Instruct. For activation/KV-cache, we utilized 8-bit assymetric per-tensor quantization scheme. Compared to the BF16 baseline, our weight-only W2 UPQ model already incurs only a modest increase in WikiText2 perplexity (10.48 → 11.49), while maintaining strong performance on CSR, MMLU, and IFEval. When we additionally quantize activations and KV-cache to INT8 (W2A8KV8), the metrics degrade only slightly (e.g., MMLU 53.20 → 52.08, IFEval 45.19 → 44.89), and remain close to the W2 case. This suggests that UPQ produces weights that are inherently robust to standard INT8 activation/KV quantization, and that full W2A8KV8 quantization is feasible with only a small loss in quality. Importantly, we achieve this without any specialized rotation-based preprocessing (e.g., SpinQuant or QuaRot), indicating that UPQ can serve as a simple and scalable backbone for end-to-end low-precision deployment.

# P EXTENDED COMPARISON OF UPQ WITH BITNET B1.58 2B4T

Table 12: Comparison of Llama 3.2 3B Instruct W2 UPQ with BitNet b1.58 2B4T.

| Method | CSR Avg. (↑) | MMLU (↑) | IFEval (↑)
(Instruct-Strict,Chat-Template) |
|---|---|---|---|
| Llama 3.2 3B Instruct | 65.44 | 59.92 | 76.86 |
| BitNet b1.58 2B4T | **68.43** | 53.17 | 53.48 |
| UPQ (Ours) | 64.28 | **55.98** | **59.11** |

Table 12 compares our W2 UPQ model on Llama 3.2 3B Instruct with BitNet b1.58 2B4T, which is a 1.58-bit model trained from scratch at a slightly smaller parameter scale. Although this is not a perfectly equal bit/memory comparison, it offers a useful reference point against a strong ultra-low-bit baseline. We observe that BitNet attains a higher CSR average (68.43 vs. 64.28), reflecting its strength as a base-language model trained end-to-end in low precision. However, UPQ achieves better performance on MMLU (55.98 vs. 53.17) and, more importantly, on IFEval (59.11 vs. 53.48), which directly measures instruction-following quality under our evaluation protocol. These results indicate that starting from a strong instruction-tuned FP16 model and applying UPQ can yield a 2-bit model that is competitive with, and in some aspects superior to, a from-scratch 1.58-bit BitNet model on knowledge- and instruction-centric benchmarks.

## Q EXTENSION OF UPQ TO QWEN MODEL

Table 13: Qwen3-4B: INT2 QAT results on various benchmarks

| Method | WikiText2 ($\downarrow$) | CSR Avg. ($\uparrow$) | MMLU ($\uparrow$) | IFEval ($\uparrow$) |
|---|---|---|---|---|
| INT4 PTQ $\rightarrow$ NTP-QAT | 8.5 | 65.0 | 50.0 | 22.0 |
| Distill-QAT | 13.0 | 61.0 | 42.5 | 30.2 |
| UPQ (Ours) | 9.8 | 64.0 | 55.1 | 44.0 |

To evaluate the generalization of UPQ toward different architecture than Llama, we applied 2-bit QAT Qwen3-4B Yang et al. (2025), which lies outside the LLaMA family. Each method was trained with 5B training tokens, and we followed same hyper-parameter configuration as Llama3.2-3B-Instruct for fair comparison.

The trends on Qwen3-4B closely mirror those observed with LLaMA-based models, confirming that our approach is not tied to the LLaMA architecture. Our UPQ method achieves the best of both worlds on Qwen3-4B. It attains near-baseline WikiText2 perplexity and CSR (only slightly higher PPL than the NTP path, with CSR almost recovered), while dramatically improving the IFEval and MMLU.

