# OpenReview forum: "Unified Progressive Quantization toward 2-bit Instruction-Tuned LLMs"
_ICLR.cc/2026/Conference — Submitted to ICLR 2026_

### Official Review · Reviewer_DqNN · 2025-10-30

**Soundness:** 1
**Presentation:** 2
**Contribution:** 1
**Rating:** 2
**Confidence:** 4

**Summary:**

The paper introduces a progressive quantization scheme designed for instruction fine-tuned LLMs. The authors claim that: (1) progressive quantization is critical to achieve 2-bit quantization of instruction fine-tuned LLMs, and (2) one needs to optimize with respect to the reference model via distillation loss rather than the next-token prediction loss. To support this, they design a 2-stage pipeline that starts with INT4 PTQ then finishes with INT2 QAT. They provide some results with WikiText2, CSR, MMLU, and IFEval.

**Strengths:**

1. Few works have explored progressive quantization, so this provides more data here.
2. The methods described in Section 3.2 to map from INT4 to INT2 are interesting. The authors provide a nice analysis and visualization.

**Weaknesses:**

1. The use of MNIST QAT as a motivating example for LLM QAT is outdated. High-accuracy, low-bit MNIST models have been available since at least 2016 [1]. For example, there is a publicly available 2-bit, 3-layer model with 64 neurons that achieves ~99% top-1 accuracy [2], substantially outperforming your illustrative baseline. This suggests the chosen motivating example does not meaningfully represent the current state of QAT.
2. The paper overlooks an important aspect of 2-bit and fewer QAT training dynamics: training from a fully converged checkpoint often underperforms compared to training from scratch or an intermediate checkpoint. Prior work such as ParetoQ explicitly analyzes this effect across initialization stages (0%–100%) and shows optimal performance at intermediate points. The omission of this consideration weakens the fairness of the comparison to ParetoQ, which is the primary baseline in the evaluation and performed best from an intermediate high-precision checkpoint.
3. The proposed theoretical framework for the quantization path is conceptually interesting but underdeveloped. It provides no clear explanation for why progressive quantization would be particularly important for instruction-tuned models, nor insight into what constitutes a “favorable” intermediate checkpoint. The claim that any intermediate 4-bit method works (L194–195) undermines the stated theoretical motivation—if simple rounding performs comparably, the framework’s added value beyond this is unclear.
4. The comparison using next-token prediction loss is not well aligned with the stated focus on instruction-tuned models. Starting QAT from an instruction fine-tuned checkpoint inherently makes next-token prediction less suitable, as it corresponds to the foundation model’s pretraining objective. A more appropriate baseline would use a post-training optimizer such as GRPO, enabling a fairer comparison against the distillation loss.
5. The evaluation and contextualization are incomplete. Prior work has demonstrated LLM quantization to 2 bits or below well before ParetoQ, including QAT (e.g., BitNet’s 1.58-bit models) and PTQ methods such as QuIP, FrameQuant, GPTAQ, Qronos, and QEP. The paper overlooks much of this literature, which narrows the perceived comparison space. While some of these techniques are contemporary, substantiating the claim that QAT is critical requires a direct, fair head-to-head comparison with recent state-of-the-art PTQ algorithms evaluated on the same calibration dataset.


References:

[1] Umuroglu et al. (2016), "FINN: A Framework for Fast, Scalable Binarized Neural Network Inference"

[2] https://github.com/fastmachinelearning/qonnx_model_zoo/tree/main/models/MNIST/Brevitas_FINN_LFC

**Questions:**

1. Can you evaluate your method against a 2-bit variant of BitNet?
2. Have you tested on other LLM architecture besides Llama?

---

> ### Author Response · Authors · 2025-11-21
> **Response to reviewer DqNN (1)**
>
> We sincerely appreciate the reviewer DqNN for valuable feedbacks,. We address the reviewer’s comments below.
>
> -------
>
> ### **Weakness 1.**
>
> The use of MNIST QAT as a motivating example for LLM QAT is outdated. High-accuracy, low-bit MNIST models have been available since at least 2016 [1]. For example, there is a publicly available 2-bit, 3-layer model with 64 neurons that achieves ~99% top-1 accuracy [2], substantially outperforming your illustrative baseline. This suggests the chosen motivating example does not meaningfully represent the current state of QAT.
>
> --------
>
> ### **Response to Weakness 1.**
>
> We agree with the reviewer that the MNIST QAT toy example is an outdated proxy for LLM quantization. While we used a vision-transformer-based MNIST experiment as an illustrative toy case, it indeed does not fully represent modern QAT for LLMs. **To address this concern, we have verified that our analysis extends to a current large language model.**
>
> In particular, **we repeated the experiment under the same setting on a LLaMA-3.2-1B Instruct model.** The results (please see Appendix N, Figure 11 in the revised manuscript) show the same trends as the toy MNIST analysis. Specifically, by reducing the weight perturbation and the maximal gradient norm during the quantization process, we tighten the upper bound of the loss variation. In turn, this yields a consistent improvement in the final model’s performance (for example, an increased MMLU score for the INT2 model) compared to a direct FP16→INT2 quantization baseline. This evidence gives us more confidence that the insights from the toy example hold for deployable LLM QAT.
>
> -------
>
> ### **Weakness 2.**
>
> The paper overlooks an important aspect of 2-bit and fewer QAT training dynamics: training from a fully converged checkpoint often underperforms compared to training from scratch or an intermediate checkpoint. Prior work such as ParetoQ explicitly analyzes this effect across initialization stages (0%–100%) and shows optimal performance at intermediate points. The omission of this consideration weakens the fairness of the comparison to ParetoQ, which is the primary baseline in the evaluation and performed best from an intermediate high-precision checkpoint.
>
> -------
>
> ### **Response to Weakness 2.**
>
> We thank the reviewer for this insightful observation. Indeed, ParetoQ (Figure 2 in that paper) found that the optimal allocation is to spend about 90% of the training budget in full-precision training and the remaining ~10% on QAT fine-tuning, rather than starting QAT from a fully converged checkpoint. Due to limited computation resources, we were regrettably unable to explore different starting points (full-precision vs. intermediate checkpoint) for UPQ in our experiments. We agree that such an analysis could further strengthen our work.
>
> That said, the context of UPQ is somewhat different from ParetoQ. **UPQ’s primary target is the post-hoc quantization of instruction-tuned LLMs** — that is, applying QAT after a model has already been fully trained (and often instruction-fine-tuned) and released as an open-source checkpoint. In this common scenario, the model is fully converged in full precision, and our goal is to compress it to 2-bit while preserving its instruction-following abilities. In other words, UPQ is meant to add value by taking a fully-trained model and making it 2-bit deployable. This is in contrast to ParetoQ’s setting, which fine-tunes a base model and could choose to stop early to do QAT.
>
> Moreover, there is a technical difference in training methodology:
>
> ParetoQ continues training on the next-token prediction objective during QAT, whereas UPQ performs knowledge distillation from a full-precision teacher model to a quantized student. For UPQ to be effective, the teacher model must have learned a good full-precision representation of the data. If we had an under-trained teacher (for example, stopping base training early to leave room for QAT), the distillation signal might be weak and the QAT could underperform. **Therefore, the strategy of using an intermediate checkpoint may not directly translate to UPQ’s distillation-based paradigm.** We will clarify this point and consider investigating intermediate-checkpoint QAT in future work.

---

> > ### Author Response · Authors · 2025-11-21
> > **Response to reviewr DqNN (2)**
> >
> > -----
> >
> > ### **Weakness 3.**
> >
> > The proposed theoretical framework for the quantization path is conceptually interesting but underdeveloped. It provides no clear explanation for why progressive quantization would be particularly important for instruction-tuned models, nor insight into what constitutes a “favorable” intermediate checkpoint. The claim that any intermediate 4-bit method works (L194–195) undermines the stated theoretical motivation—if simple rounding performs comparably, the framework’s added value beyond this is unclear.
> >
> > ------
> >
> > ### **Response to Weakness 3.**
> >
> > We appreciate the request for a clearer theoretical motivation regarding favorable intermediate checkpoints. We acknowledge that our original claim was overstated: it is not that any INT4 initialization method will automatically yield benefit. We have revised the manuscript to better articulate why a well-chosen intermediate quantization (e.g., 4-bit PTQ) helps.
> >
> > Our theoretical framework (Equation (3) in the manuscript) suggests that the loss deviation during quantization is bounded in part by a weight perturbation term. The smaller this perturbation, the tighter the bound on loss difference. Therefore, an intermediate checkpoint is “favorable” if it significantly reduces the weight perturbation incurred in the final 2-bit quantization step. The question becomes: How does using an INT4 PTQ initializer actually reduce the weight perturbation for the subsequent INT2 quantization? We provide two key analytical insights in Appendix M of the revised manuscript to answer this:
> >
> > * **Outer-bin occupancy reduces quantization error** : Using an INT4 initialization increases the fraction of weights that end up in the tail quantization bins of the final INT2 quantizer. Intuitively, if more weights saturate at the ±max values of the 2-bit quantizer, the error per weight is limited by that saturation (since further increases in those weights do not increase quantization error). Our analysis shows that increasing this outer-bin occupancy strictly reduces the fixed-$\alpha$ quantization error for 2-bit weights.
> >
> > * **Range reduction yields quadratic error shrinkage** : The INT4 PTQ step compresses the weight distribution’s range before we do INT2 quantization. We show that this range reduction leads to a much smaller quantization error when going to 2 bits — in fact, the error shrinks approximately quadratically compared to skipping the INT4 step. Thus, an INT4 initializer provides a significantly better starting point by ensuring the INT2 quantizer is only dealing with a narrow, well-behaved weight range.
> >
> > These insights shed light on why progressive quantization is particularly helpful: instruction-tuned models often have outlier weights or wider weight distributions that are hard to quantize directly to 2-bit without incurring large error. A 4-bit PTQ intermediate checkpoint alleviates this by (a) pushing more weights into stable saturation zones and (b) tightening the overall range of weights, both of which make the final 2-bit quantization more accurate.
> >
> > Empirically, we see evidence of these effects in our weight distribution plots. For example, Figure 4 and Figure 9 in the manuscript show that after using an INT4 PTQ step, the weight distribution of the model becomes narrower and has a higher concentration of weights at the extreme bins, compared to the direct FP16→INT2 case.

---

> ### Author Response · Authors · 2025-11-21
> **Response to reviewr DqNN (3)**
>
> -----
>
> ### **Weakness 4.**
>
> The comparison using next-token prediction loss is not well aligned with the stated focus on instruction-tuned models. Starting QAT from an instruction fine-tuned checkpoint inherently makes next-token prediction less suitable, as it corresponds to the foundation model’s pretraining objective. A more appropriate baseline would use a post-training optimizer such as GRPO, enabling a fairer comparison against the distillation loss.
>
> -----
>
> ### **Response to Weakness 4.**
>
> We agree that exploring post-training optimization objectives (such as alignment tuning methods) for QAT could be a valuable direction. In fact, one could imagine replacing or augmenting our distillation loss with techniques like GRPO or DPO to better align the quantized model with instruction-following tasks. We have not yet explored these avenues, and we acknowledge this as an interesting area for future work.
>
> However, we note some practical challenges with such approaches in the QAT setting:
>
> * **On-policy RL-style methods (e.g. GRPO)** require the model to generate its own outputs during training. In a model with 2-bit quantizer attached, generation from the LLM is extremely slow, which would make on-policy training prohibitively time-consuming for models of this size. When we enable on-policy generation during training, the computation time increases about 10X than off-policy training.
>
> * **Preference-based tuning methods (e.g. DPO)** avoid on-policy generation but require a substantial dataset of human preference labels or comparison data. Running a full DPO or similar RLHF-style pipeline at the scale of our QAT (we use on the order of 30B tokens) is not trivial and would demand significant additional data and computing resources beyond our current scope.
>
> In contrast, our chosen approach — knowledge distillation from the original instruction-tuned FP16 model — is highly scalable because it uses the same unlabeled corpora and teacher-student framework without needing human feedback or on-the-fly generation. Given the scalability issues noted above, we believe our distillation strategy was a reasonable and effective choice for preserving the instruction-tuned capabilities of the quantized model.
>
> -----

---

> ### Author Response · Authors · 2025-11-21
> **Response to reviewer DqNN (4)**
>
> -------
>
> ### **Weakness 5.**
>
> The evaluation and contextualization are incomplete. Prior work has demonstrated LLM quantization to 2 bits or below well before ParetoQ, including QAT (e.g., BitNet’s 1.58-bit models) and PTQ methods such as QuIP, FrameQuant, GPTAQ, Qronos, and QEP. The paper overlooks much of this literature, which narrows the perceived comparison space. While some of these techniques are contemporary, substantiating the claim that QAT is critical requires a direct, fair head-to-head comparison with recent state-of-the-art PTQ algorithms evaluated on the same calibration dataset.
>
>  -----
>
> ### **Response to Weakness 5.**
>
> We agree that comparing against a broad range of prior ultra-low-bit quantization methods (both PTQ and QAT) is important to put our contributions in context. Below, we also provide additional comparisons and clarifications:
>
> * **Comparison to recent PTQ methods** : Many of the cited PTQ methods use different quantization granularities and strategies, which makes a perfectly fair “apple-to-apple” comparison challenging. For example, QuIP# uses vector quantization (with an 8-dimensional lattice codebook), Qronos and GPTAQ use asymmetric per-channel quantization. For clarity, we summarize some key characteristics of these methods versus UPQ in the table below:
>
> | Method      | 2-bit Support         | Quantization Granularity              | Quantization Function                          | Coherent Processing |
> |------------|------------------------|---------------------------------------|------------------------------------------------|-------------------------------|
> | QuIP#   [5]   | ✓      | Vector quant. (8-D block)             | E8 lattice VQ (non-uniform)                    | ✓                             |
> | Qronos [2]    | ✓                      | Per-channel                           | Asymmetric uniform per-channel                 |          △                    |
> | QEP   [3]    | ✓                      | Per-channel or group-wise (e.g., 32)  | Symmetric uniform per-channel                  | ✗                             |
> | GPTAQ [1]      | ✓                      | Per-channel                           | Asymmetric uniform per-channel                 | ✗                             |
> | UPQ (ours) | ✓                      | Per-channel                           | Symmetric uniform per-channel (SEQ)      | ✗                             |
>
> Each of these methods has different goals and setups, so achieving a unified comparison on one benchmark requires aligning many details (e.g., using the same quantization grid, same grouping, etc.). Nonetheless, to gauge the benefit of QAT over PTQ, we did compare UPQ with some available PTQ baselines under similar conditions.
>
> **WikiText-2 perplexity (↓) for QAT-based UPQ and PTQ baselines**
>
> | Method                    | LLaMA-3.2-1B | LLaMA-3.2-3B | LLaMA-3.1-8B |
> |---------------------------|-------------:|-------------:|-------------:|
> | FP (Full-Precision)       | **12.14**    | **10.48**    | **6.75**     |
> | **UPQ (Ours, QAT)**       | 15.46        | 11.49        | 8.42         |
> | Qronos [2] (PTQ)          | 24.60        | 14.90        | 12.40        |
> | QuaRot + GPTAQ [1] (PTQ)  | –            | –            | 13.40        |
> | QuIP [4] (PTQ)     | –            | -       | 70.518           |
> | QuIP + QEP [3] (PTQ) | –         | -      | 27.326           |
>
> Our QAT-based UPQ achieves perplexity very close to the full-precision baseline (e.g. 11.49 vs 10.48 for the 3B model, a small difference). In contrast, purely PTQ methods suffer significant degradation at the same bit-width. For example, Qronos yields 14.9 perplexity much higher (worse) than the FP baseline’s 10.48. This quantifies that the QAT step significantly closes the gap between 2-bit quantized models and the FP model.
>
> We focused on WikiText-2 perplexity for this comparison because most PTQ methods do not report performance on more complex tasks or benchmarks, making direct comparisons difficult. Additionally, note that some of the better-performing “PTQ” methods are not truly stand-alone PTQ techniques. For example, Qronos requires extra quantization transformations before applying its rounding algorithm, and QEP is an error-correction module applied on top of a layer-wise PTQ method (as in QuIP + QEP). These extra steps go beyond standard post-training quantization, underscoring that achieving strong results often requires techniques beyond a basic PTQ pipeline.

---

> ### Author Response · Authors · 2025-11-21
> **Response to reviewer DqNN (5)**
>
> ### **Response to Weakness 5. (Continued)**
>
> * **Comparison to BitNet** : We also note that BitNet, while related as low-bit quantization methods, has different problem settings compared to UPQ.
>
> BitNet trains models from scratch with ultra-low precision weights. It achieved impressive results with a ternary (1.58-bit) weight quantizer (using a BitLinear quantization scheme) on base LLMs. However, BitNet’s scenario is to train a new model from the ground up in low precision; it does not directly apply to compressing an already instruction-tuned model without retraining from scratch.
>
> These distinctions are summarized in the table below:
>
> | Method    | Training                           | Targets Base LLM? | Targets Instruct LLM? | Bit-width & Quantizer                       |
> |-----------|--------------------------------------------|-------------------|-----------------------|---------------------------------------------|
> | ParetoQ   | QAT with pre-trained model (≈90% FP + 10% QAT)  | ✓ (base models)   | ✗ (not specifically)  | 2-bit QAT, SEQ (stretched elastic)          |
> | BitNet    | Train from scratch in low-bit              | ✓ (base models)   | ✗ (not applicable)    | 1.58-bit (ternary) BitLinear quantizer      |
> | UPQ (ours)|  QAT on an instruction-tuned model    | ✗ (starts from tuned model) | ✓ (instruction-tuned models) | 2-bit QAT, SEQ (stretched elastic) |
>
> Because of these different settings, a direct head-to-head comparison is not straightforward. For example, evaluating UPQ against a “2-bit BitNet” would require a BitNet model checkpoint at 2-bit precision, which is not available as an open-source. Instead, we compared our method with the closest available BitNet model in terms of model size. We took the open-source BitNet model (1.58-bit weights, roughly 2B parameters) and compared it to a LLaMA-3.2B-Instruct model quantized to 2-bit with UPQ. The results are as follows:
>
> | Method                                 | CSR Avg ↑ | MMLU ↑ | IFEval ↑ *(Instr-Strict, Chat-Template)* |
> |----------------------------------------|----------:|-------:|------------------------------------------:|
> | LLaMA-3.2-3B-Instruct (FP16 baseline)  | 65.44     | 59.92  | 76.86                                     |
> | BitNet (1.58-bit, ~2B params, scratch) | **68.43** | 53.17  | 53.48                                     |
> | **UPQ 2-bit (LLaMA-3.2-3B-Instruct)**  | 64.28     | **55.98** | **59.11**                                |
>
> Although this is not a perfectly equal comparison in terms of model size or bit-width (BitNet uses ~1.58 bits and has slightly fewer effective parameters), it provides some insight. We see that BitNet (trained from scratch) attains a higher CSR average score, likely because it was optimized for that pretraining-style metric. However, our 2-bit UPQ model achieves better performance on knowledge-intensive and instruction tasks (MMLU and IFEval). This suggests that starting from a strong instruction-tuned base and applying UPQ is a viable strategy to preserve complex abilities in the low-bit scheme.
>
> -----
>
> ### **Question 1.**
>
> Can you evaluate your method against a 2-bit variant of BitNet?
>
> -----
>
>
> ### **Response to Question 1.**
>
> Please see Response to Weakness 5.
>
> -----
>
>
> ### **Question 2.**
>
> Have you tested on other LLM architecture besides Llama?
>
> -----
>
> ### **Response to Question 2.**
>
> We agree that evaluating on a broader range of models strengthens the claims of the paper. To demonstrate that our approach is not limited to the LLaMA family, we are extending our experiments to Qwen3-4B, which is outside of Llama family.
>
> As the whole experiments are not finished, we will remind reviewer DqNN about these results during the rebuttal timeframe.
>
> ------
>
> [1] Li, Yuhang, et al. "GPTAQ: Efficient Finetuning-Free Quantization for Asymmetric Calibration." Forty-second International Conference on Machine Learning.
>
> [2] Zhang, Shihao, et al. "Qronos: Correcting the Past by Shaping the Future... in Post-Training Quantization." arXiv preprint arXiv:2505.11695 (2025).
>
> [3] Arai, Yamato, and Yuma Ichikawa. "Quantization Error Propagation: Revisiting Layer-Wise Post-Training Quantization." arXiv preprint arXiv:2504.09629 (2025).
>
> [4] Chee, Jerry, et al. "Quip: 2-bit quantization of large language models with guarantees." Advances in Neural Information Processing Systems 36 (2023): 4396-4429.
>
> [5] Tseng, Albert, et al. "Quip#: Even better llm quantization with hadamard incoherence and lattice codebooks." arXiv preprint arXiv:2402.04396 (2024).
>
> Once again, we sincerely thank the reviewer for the thoughtful and constructive feedback. We hope our responses and the revised experiments address the concerns, and we are happy to clarify any remaining questions.

---

> > ### Comment · Reviewer_DqNN · 2025-11-25
> >
> > Thank you for your detailed responses. After reviewing them carefully, I believe the paper still requires additional evidence and analysis before it can be considered for acceptance. While the idea of progressive quantization is interesting, the current empirical and theoretical support is not sufficient to substantiate the core claims. Below, I summarize my remaining concerns.
> >
> > **Novelty and Positioning:** The distinction from ParetoQ is framed around (a) progressive quantization and (b) distillation-based QAT. However, distillation-based QAT is well-established (even for LLMs) and not novel (see [1,2] as two examples).
> >
> > **Empirical Evidence:** The claim that progressive quantization is critical for quantizing instruction-tuned models remains insufficiently supported. While compute limitations are understandable, such a strong claim requires stronger empirical validation than what is presented. The comparison against PTQ is acknowledged and appreciated (though the analysis does not appear in the latest revision), but stronger QAT baselines are still missing. For example, the ParetoQ framework appears model-, dataset-, and loss-agnostic, suggesting it could trivially be extended to post-training scenarios. A simple baseline could involve applying a state-of-the-art post-training optimizer (e.g., GRPO or DPO), then introducing SEQ quantizers for a short QAT phase (e.g., 10%) to reach high-quality 2-bit instruction fine-tuned models. This would help test whether progressive quantization is truly necessary compared to hybrid strategies. Currently, this evaluation (or a similar one) is missing.
> >
> > **Theoretical Analysis:** The theoretical component still feels underdeveloped. For instance, Equation 3 provides only a weak upper bound. It still provides no clear explanation for why progressive quantization would be particularly important for instruction-tuned models, nor insight into what constitutes a “favorable” intermediate checkpoint. Without rigorous empirical baselines, the conclusions drawn from the UPQ framework are not convincing.
> >
> > In summary, the paper introduces progressive quantization as a simple and potentially useful idea, but the lack of rigorous empirical validation and theoretical differentiation makes it premature for acceptance. Strengthening the experimental section with hybrid baselines and refining the theoretical analysis would significantly improve the work.
> >
> > References:
> > 1. BitDistiller: Unleashing the Potential of Sub-4-Bit LLMs via Self-Distillation
> > 2. LLM-QAT: Data-Free Quantization Aware Training for Large Language Models

---

> > > ### Author Response · Authors · 2025-12-04
> > > **Response to Reviewer DqNN (final to Area Chair)**
> > >
> > > We thank Reviewer DqNN for their very detailed and technically sophisticated review. We understand that they remain unconvinced after our initial rebuttal, but we respectfully disagree with several aspects of their final assessment, and we want to clearly explain this for the Area Chair.
> > >
> > > ------
> > >
> > > ### **(1) On novelty and positioning**
> > >
> > > Reviewer 4 argues that our distinction from ParetoQ is insufficient because distillation‑based QAT is not new. We want to clarify our claims.
> > >
> > > - A unified progressive pipeline specialized for instruction‑tuned LLMs: FP16 → **INT4 PTQ (zero‑free SEQ‑aligned grid)** → INT2 QAT, with a single FP16 instruction‑tuned teacher.
> > > - A quantization‑path analysis (Eq. 3 and Appendices M,N) that links the progressive path to a loss bound, and identifies specific mechanisms—outer‑bin occupancy and range reduction—by which INT4 PTQ improves the subsequent INT2 quantization.
> > > - A systematic study of 2‑bit quantization for instruction‑tuned models under a realistic post‑hoc quantization scenario where one only has access to public pretraining corpora, not the original SFT/RL data.
> > >
> > > Distillation QAT itself is indeed established, but **its combination with a carefully designed INT4 PTQ stage, analyzed both theoretically and empirically for instruction‑tuned model** is new and directly addresses failure modes that ParetoQ experiences on IFEval and MMLU when applied to such models.
> > >
> > > ---------
> > > ### **(2) On empirical evidence and missing baselines**
> > >
> > > Reviewer 4 requests stronger “hybrid” baselines (e.g. ParetoQ + GRPO/DPO) and suggests that ParetoQ could trivially be extended to post‑training scenarios. We argue that, in our target setting, such baselines are not realistic and do not constitute a fair comparison:
> > >
> > > **Problem definition difference.**
> > > UPQ is designed for post‑hoc compression of already instruction‑tuned models when the original SFT/RL data are unavailable—a common situation for open‑source deployments. In our setting, we assume access only to public, general‑domain corpora (e.g., DCLM‑Edu) and the FP16 teacher; we do not assume a large preference dataset or the ability to re‑run a full RLHF pipeline. Requiring us to gather and use a massive human‑preference dataset to run GRPO/DPO fundamentally changes the problem being solved.
> > >
> > > **Computational feasibility at 2‑bit QAT.**
> > > On‑policy RL‑style methods like GRPO require repeated sampling from the student model (quantized model) during training. This generation requires further computation time, so A GRPO‑style QAT phase on top of 30B tokens would be prohibitively expensive for 3B/8B sized models under the same resource budget.
> > >
> > > **Off‑policy preference optimization (e.g., DPO) still needs large preference datasets.**
> > > DPO avoids on‑policy generations but relies on enough size of human preference datasets. Such datasets are not publicly available at the scale needed to match our QAT budget, and creating them would again step outside the post‑hoc compression problem we explicitly target.
> > >
> > > For these reasons, we believe comparing UPQ against “ParetoQ + GRPO/DPO” is not an apples‑to‑apples baseline, but rather a different research problem: re‑training a 2‑bit aligned model from scratch or with a new, large preference dataset.
> > >
> > > -------
> > >
> > > ### **(3) On the theoretical analysis**
> > > Reviewer 4 characterizes our theory as a weak upper bound that does not explain favorable checkpoints. We agree that Eq. 3 alone is a relatively loose bound, but the revision goes significantly further:
> > >
> > > - Appendix M provides a specific condition where INT4 PTQ reduces INT2 error via:
> > >   - increased outer‑bin occupancy (more weights mapping into ±3 leads to strictly reduced fixed‑α error), and
> > >   - range reduction that yields a quadratic κ² shrinkage of INT2 error when INT4 effectively rescales the weight range by κ<1.
> > >
> > > - Appendix N validates these mechanisms on a real 1B‑parameter LLM: along the INT4→INT2 path, both ΔW and the sampled $G_\text{max}$ are smaller, and Δ$W$·$G_\text{max}$ tracks the improved MMLU scores during QAT.
> > > Thus, while we do not claim a theoretical background on every possible progressive quantization, we do provide both analytical and empirical evidence that the INT4 PTQ checkpoint used by UPQ is “favorable” in the sense of reducing the relevant bound and improving downstream performance.

---

> > > > ### Author Response · Authors · 2025-12-04
> > > > **Response to Reviewer DqNN (final to Area Chair)**
> > > >
> > > > -----------
> > > >
> > > > ### **(4) Broader evaluation beyond the LLaMA family (Question 2)**
> > > > Reviewer DqNN requested evaluation on a non‑LLaMA architecture. During the rebuttal period, we completed UPQ on Qwen3‑4B which lies outside the LLaMA family.
> > > > Each method was trained with 5B training tokens, and we followed same hyper-parameter configuration as Llama3.2-3B-Instruct for fair comparison. We applied the same three quantization setups (INT4 PTQ → NTP-QAT, Distill-QAT, and UPQ) and measured performance on WikiText2 (perplexity), CSR (common sense reasoning average score), MMLU, and IFEval.
> > > >
> > > > | Method | WikiText2 (↓) | CSR Avg. (↑) | MMLU (↑) | IFEval (↑) |
> > > > |---|---:|---:|---:|---:|
> > > > | INT4 PTQ → NTP-QAT | 8.5 | 65.0 | 50.0% | 22.0% |
> > > > | Distill-QAT | 13.0 | 61.0 | 42.5% | 30.2% |
> > > > | UPQ (Ours) | 9.8 | 64.0 | 55.1% | 44.0% |
> > > >
> > > > The trends on Qwen3‑4B closely mirror those observed with LLaMA-based models, confirming that our approach is not tied to the LLaMA architecture. Our UPQ method achieves the best of both worlds on Qwen3-4B. It attains near-baseline WikiText2 perplexity and CSR (only slightly higher PPL than the NTP path, with CSR almost recovered), while dramatically improving the instruction and knowledge metrics.
> > > >
> > > > -----------
> > > > ### **Summary of disagreement**
> > > > In summary, we respectfully disagree with Reviewer DqNN’s conclusion that the work is premature. Since the initial submission we have:
> > > >
> > > > - Added substantial theoretical development (Appendices M,N) linking the INT4 intermediate to quantization error and observed training behavior.
> > > > - Extended experiments to a non‑LLaMA model (Qwen3‑4B), showing the same qualitative pattern.
> > > > - Included comparisons with multiple state‑of‑the‑art 2‑bit PTQ methods, demonstrating that QAT (UPQ) is still essential at this precision.
> > > > - Explored multiple losses, datasets, grids, and intermediate bit‑widths, positioning UPQ as a general progressive pipeline.

---

### Official Review · Reviewer_6U4g · 2025-10-31

**Soundness:** 2
**Presentation:** 2
**Contribution:** 2
**Rating:** 4
**Confidence:** 2

**Summary:**

This paper proposes a pipeline: FP16-> INT4 -> INT2, where the first stage uses PTQ, and the second stage uses QAT. They design distillation-based QAT Distill-QAT using JSD loss instead of the next-token prediction loss. Ablations show strong sensitivity to intermediate bitwidth, distillation objectives (GJS vs NTP).

**Strengths:**

1. The paper is easy to follow.
2. The ablations are insightful.

**Weaknesses:**

1. The motivation is not clear. Using INT4 PTQ as a good initializer does not guarantee that $\Delta W$ would decrease.
2. Table 2-5 should also report standard deviation.
3. The evaluations are within the Llama family. Broader evaluation should enhance the claim.
4. Lack of baselines for lower-precision PTQ methods. The author needs to quantify: how much gain does the QAT step actually add?
5. This paper's current scope is rather narrow, focusing too much on specific steps. Given its high sensitivity to intermediate bitwidth and distillation objectives, the reviewer recommends that this paper be reformulated as a general, progressive low-precision pipeline, and evaluating multiple intermediate bitwidths and target bitwidths.

**Questions:**

Will co-distillation from multiple higher-bitwidth models, such as 4-bit,8-bit models obtained via PTQ, further enhance the performance?

---

> ### Author Response · Authors · 2025-11-21
> **Response to Reviewer 6U4g (1)**
>
> We sincerely appreciate the reviewer 6U4g for valuable feedbacks and recognition of insightful ablations and easy-to-follow logics of our paper. We address the reviewer’s comments below.
>
> ---------------------------
>
> ### **Weakness 1.**
>
> The motivation is not clear. Using INT4 PTQ as a good initializer does not guarantee that weight  would decrease.
>
>  ---------
>
> ### **Response to Weakness 1.**
>
> We thank the reviewer for pointing out this concern. In Figure 3 of our manuscript, we observed that after applying INT4 post-training quantization (PTQ), the magnitude of weight perturbation is significantly reduced, and this trend remains stable throughout further training. This empirical observation serves as evidence supporting our approach’s motivation.
>
> However, we agree that this experimental finding alone does not guarantee a reduction in weight perturbation during the subsequent INT2 quantization stage in all cases. **To strengthen our claims, we have added a more analytical investigation in Appendix M of the revised manuscript.**
>
> Analytically, our analysis in Appendix M provides two key insights into why an INT4 initializer helps reducing weight perturbation on INT2 quantization:
>
> * **Outer-bin occupancy reduces quantization error** : Increasing the proportion of weights that occupy the outermost bins (i.e. weights with $|u| > 2\alpha$) in the INT2 quantizer strictly reduces the fixed-$\alpha$ quantization error . Intuitively, more weights hitting the saturation threshold can limit the error per weight, and this effect can outweigh the small Lipschitz slack (residual error from clipping) in the quantization process.
>
> * **Range reduction yields quadratic error shrinkage** : Using INT4 PTQ as an initialization shrinks the INT2 quantization error in a quadratic manner. In other words, the INT4 step compresses the weight range, so when we further quantize to 2-bit, the error is much smaller than it would be without the INT4 step. This analysis explains the strong improvement observed when handing off from INT4 to INT2 (as illustrated by the weight distribution in Figure 2 of our paper).
>
> **Figure 4 and Figure 9 of the manuscript show that, unlike the direct FP16→INT2 case, using an INT4 PTQ step produces a weight distribution with a narrower range and a higher concentration of weights in the OUTER quantization bins.** According to the above analysis, both of these effects (narrower range and outer-bin occupancy) drive the INT2 quantization error lower. Furthermore, Figure 9 demonstrates that even in a non-distillation setting (using an NTP-QAT approach), we observe the same trend: the INT4-initialized model has a weight distribution that leads to significantly less INT2 perturbation than a model quantized directly from FP16.
>
> -------
>
> ### **Weakness 2.**
>
> Table 2-5 should also report standard deviation.
>
> --------
>
> ### **Response to Weakness 2.**
>
> We acknowledge the importance of reporting variability. Unfortunately, due to computational constraints, we were only able to run a single training for each configuration in our experiments. This means we do not have multiple runs per setting to compute standard deviations for Tables 2–5. We recognize that the absence of standard deviation makes it harder to assess the reliability of the results, and we apologize for this limitation. In the final manuscript, we will include standard deviations for as many results as possible.
>
> We also want to emphasize that the major performance trends were consistent across all our experiments, which gives us confidence in the significance of the results despite the lack of multiple trials. For example, we consistently observed the IFEval metric collapsing under the NTP-QAT baseline, whereas it was recovered and improved by our UPQ method in every case. Such repeatable patterns suggest that the improvements introduced by UPQ are robust, even if we could not quantify the variance in every table.
>
> ------
>
> ### **Weakness 3.**
>
> The evaluations are within the Llama family. Broader evaluation should enhance the claim.
>
> ------
>
> ### **Response to Weakness 3.**
>
> We agree that evaluating on a broader range of models strengthens the claims of the paper. To demonstrate that our approach is not limited to the LLaMA family, we are extending our experiments to Qwen3-4B, which is outside of Llama family.
>
> **As the whole experiments are not finished, we will remind reviewer 6U4g about these results during the rebuttal timeframe.**

---

> ### Author Response · Authors · 2025-11-21
> **Response to Reviewer 6U4g (2)**
>
> -----
>
> ### **Weakness 4.**
>
> Lack of baselines for lower-precision PTQ methods. The author needs to quantify: how much gain does the QAT step actually add?
>
>  -----
>
> ### **Response to Weakness 4.**
>
> We have collected comparable perplexity results on the WikiText-2 dataset for our method and several PTQ baselines to quantify the benefit of the QAT step.  We surveyed several of the most recent post-training quantization methods which support 2-bit, and a concise summary of the key comparison is given in the table below.
>
> | Method      | 2-bit Support         | Quantization Granularity              | Quantization Function                          | Coherent Processing |
> |------------|------------------------|---------------------------------------|------------------------------------------------|-------------------------------|
> | QuIP#   [5]   | ✓      | Vector quant. (8-D block)             | E8 lattice VQ (non-uniform)                    | ✓                             |
> | Qronos [2]    | ✓                      | Per-channel                           | Asymmetric uniform per-channel                 | ✗                             |
> | QEP   [3]    | ✓                      | Per-channel or group-wise (e.g., 32)  | Symmetric uniform per-channel                  | ✗                             |
> | GPTAQ [1]      | ✓                      | Per-channel                           | Asymmetric uniform per-channel                 | ✗                             |
> | UPQ (ours) | ✓                      | Per-channel                           | Symmetric uniform per-channel (SEQ)      | ✗                             |
>
> **Many of the PTQ methods use different quantization granularities and strategies, which makes a perfectly fair “apple-to-apple” comparison challenging.** For example, QuIP# uses vector quantization (with an 8-dimensional lattice codebook), Qronos and GPTAQ use asymmetric per-channel quantization. Nonetheless, to gauge the benefit of QAT over PTQ, we did compare UPQ with some available PTQ baselines under similar conditions.
>
> | Method                    | LLaMA-3.2-1B | LLaMA-3.2-3B | LLaMA-3.2-8B |
> |---------------------------|-------------:|-------------:|-------------:|
> | FP      | **12.14**    | **10.48**    | **6.75**     |
> | **UPQ (Ours, QAT)**       | 15.46        | 11.49        | 8.42         |
> | Qronos [2] (PTQ)          | 24.60        | 14.90        | 12.40        |
> | QuaRot + GPTAQ [1] (PTQ)  | –            | –            | 13.40        |
> | QuIP [4] (PTQ)     | –            | –       | 70.518            |
> | QuIP + QEP [3] (PTQ) | –         | –       | 27.326            |
>
> **Our QAT-based UPQ achieves perplexity very close to the full-precision baseline (e.g. 11.49 vs 10.48 for the 3B model, a small difference). In contrast, purely PTQ methods suffer significant degradation at the same bit-width.** For example, Qronos yields 14.9 perplexity much higher (worse) than the FP baseline’s 10.48.
>
> Comparing these, we see the QAT step provides dramatic improvements in performance. The QAT-enhanced model’s perplexity is only ~10% worse than FP, whereas PTQ-only can be 40%–>6× worse. This quantifies that the QAT step significantly closes the gap between 2-bit quantized models and the FP model.
>
> We focused on WikiText-2 perplexity for this comparison because most PTQ methods do not report performance on more complex tasks or benchmarks, making direct comparisons difficult. Additionally, note that some of the better-performing PTQ methods are not truly stand-alone PTQ techniques. For example, Qronos requires extra quantization transformations before applying its rounding algorithm, and QEP is an error-correction module applied on top of a layer-wise PTQ method (as in QuIP + QEP). These extra steps go beyond standard post-training quantization, underscoring that achieving strong results often requires techniques beyond a basic PTQ pipeline.
>
> -----------
>
> [1] Li, Yuhang, et al. "GPTAQ: Efficient Finetuning-Free Quantization for Asymmetric Calibration." Forty-second International Conference on Machine Learning.
>
> [2] Zhang, Shihao, et al. "Qronos: Correcting the Past by Shaping the Future... in Post-Training Quantization." arXiv preprint arXiv:2505.11695 (2025).
>
> [3] Arai, Yamato, and Yuma Ichikawa. "Quantization Error Propagation: Revisiting Layer-Wise Post-Training Quantization." arXiv preprint arXiv:2504.09629 (2025).
>
> [4] Chee, Jerry, et al. "Quip: 2-bit quantization of large language models with guarantees." Advances in Neural Information Processing Systems 36 (2023): 4396-4429.
>
> [5] Tseng, Albert, et al. "Quip#: Even better llm quantization with hadamard incoherence and lattice codebooks." arXiv preprint arXiv:2402.04396 (2024).

---

> ### Author Response · Authors · 2025-11-21
> **Response to Reviewer 6U4g (3)**
>
> ------
>
> ### **Weakness 5.**
>
> This paper's current scope is rather narrow, focusing too much on specific steps. Given its high sensitivity to intermediate bitwidth and distillation objectives, the reviewer recommends that this paper be reformulated as a general, progressive low-precision pipeline, and evaluating multiple intermediate bitwidths and target bitwidths.
>
> -----
>
> ### **Response to Weakness 5.**
>
> We appreciate the reviewer’s suggestion to frame our work in a more general context. In the final manuscript, we will explicitly present our approach as a general progressive QAT pipeline, rather than a set of isolated techniques. To support this broader positioning, we have conducted extensive ablation studies that explore multiple intermediate bitwidths and other design choices. These studies demonstrate that our approach is not tied to a singular configuration (e.g., INT16→INT4→INT2 with one specific loss), but is a flexible pipeline.
>
> In particular, we investigated:
>
> * **Intermediate Bitwidth Choices (Table 3)** : We experimented with various intermediate precision levels (beyond just 4-bit) on the path to 2-bit quantization. The results show how different intermediate bitwidths affect the final INT2 performance, providing insight into the optimal setting and confirming that the progressive strategy holds general value
>
> * **Quantization Grid for 2-bit Weights (Table 2)**: We tried different quantization grid designs for the 2-bit weights (for example, adjusting the quantization scale or zero-point). It indicates that our improvements are not dependent on a particular quantization scheme or grid—further evidence of the framework’s robustness.
>
> * **Data Selection for QAT (Table 4)**: We varied the fine-tuning data used during the distillation (e.g., trying alternative datasets or subsets) to see how it influences outcomes. This addresses the concern that our results might be narrowly tuned to a particular data setting.
>
> * **Distillation Objective Variants (Table 5)**: We evaluated different loss functions and training objectives for the QAT step (for instance, comparing our guided knowledge distillation loss against standard distillation or other custom losses). The findings show that UPQ can integrate various objectives effectively.
>
> Together, these ablations paint a comprehensive picture of a general progressive quantization pipeline. We now better understand how intermediate bitwidths and other components affect the final outcome, and we show that each component of UPQ (the PTQ step, the intermediate precision, the data, and the loss) can be adjusted orthogonally without breaking the overall performance gains.
>
> In the final manuscript, we will make it explicit that our method is capable of incorporating multiple intermediate bitwidths and tuning different training objectives — rather than a fixed sequence of steps.
>
> ---------
>
> ### **Question 1.**
>
> Will co-distillation from multiple higher-bitwidth models, such as 4-bit,8-bit models obtained via PTQ, further enhance the performance?
>
> -----
>
> ### **Response to Question 1.**
>
> We appreciate reviewer’s suggestion on distillation for further improvements. Having said that, there are two well‑established findings on weak teacher.
>
> * [6] experimentally found out that distilling from weaker/biased teachers can hurt or give diminishing returns; several studies document that as teacher quality drops, student quality can degrade or plateau even with careful loss weighting. Mixing in quantized teachers (which slightly distort logits and calibration) can therefore inject noise into the target distribution.
>
> * Also, [7] states that the benefit of KD correlates with well‑calibrated teacher probabilities; perturbations from PTQ could harm transfer unless explicitly corrected.
>
> Given these findings, If we replace it with a PTQ teacher, the student is driven toward a distribution where the PTQ teacher may be (i) noisier in the tails and (ii) slightly mis‑calibrated around decision boundaries.
>
> [6] Lee, Hayeon, et al. "A Study on Knowledge Distillation from Weak Teacher for Scaling Up Pre-trained Language Models." Findings of the Association for Computational Linguistics: ACL 2023. 2023.
>
> [7] Guo, Z., Wang, D., He, Q. et al. Leveraging logit uncertainty for better knowledge distillation. Sci Rep 14, 31249 (2024).
>
> ----------
>
> Once again, we sincerely thank the reviewer for the thoughtful and constructive feedback. We hope our responses and the revised experiments address the concerns, and we are happy to clarify any remaining questions.

---

> > ### Author Response · Authors · 2025-12-04
> > **Response to Reviewer 6U4g (final summary, to Area Chair)**
> >
> > Although Reviewer 6U4g did not update their score after rebuttal, we took their comments very seriously and made substantial theoretical and empirical revisions in rebuttal response. Here we summarize how each concern is addressed, and we also report the new non‑LLaMA results that were requested.
> >
> > ------------------
> >
> > ### **(1) Clarifying the motivation and theory behind INT4→INT2 (Weakness 1)**
> > Reviewer 3 questioned whether an INT4 PTQ initializer truly reduces INT2 perturbation and asked for a clearer theoretical motivation. In the revised manuscript,
> >
> > - We added Appendix M, which analytically studies when an intermediate INT4 step strictly reduces the INT2 error, introducing:
> >   - an outer‑bin occupancy effect—moving more weights into the outer 2‑bit bins {±3} strictly reduces fixed‑α INT2 quantization error; and
> >   - a range‑reduction effect—INT4 PTQ effectively scales the weight range by κ<1, and the subsequent INT2 error shrinks approximately as κ².
> >
> > - Extended the toy MNIST analysis to a real LLM (Llama‑3.2‑1B‑Instruct) in Appendix N, showing that the ΔW·G_max bound is consistently tighter along the INT4→INT2 path than along the direct FP16→INT2 path, and that this correlates with higher MMLU throughout training (Figure 11).
> >
> > Together, these additions provide a much more concrete explanation of why a progressive path is favorable for instruction‑tuned models.
> >
> > ------------------
> >
> > ### **(2) Standard deviations and experimental robustness (Weakness 2)**
> > Reviewer 3 asked for standard deviations on Tables 2–5. Our compute budget still does not allow full multi‑seed runs for every 2‑bit experiment. But it should be noted that our UPQ performs largely better than other baselines. Having said that, we are transparent that LLM results remain single‑seed due to resource constraints.
> >
> > ------------------
> >
> > ### **(3) Broader evaluation beyond the LLaMA family (Weakness 3, Question 2)**
> > Reviewer 3 and 4 requested evaluation on a non‑LLaMA architecture. During the rebuttal period, we completed UPQ on Qwen3‑4B which lies outside the LLaMA family. Each method was trained with 5B training tokens, and we followed same hyper-parameter configuration as Llama3.2-3B-Instruct for fair comparison. We applied the same three quantization setups (INT4 PTQ → NTP-QAT, Distill-QAT, and UPQ) and measured performance on WikiText2 (perplexity), CSR (common sense reasoning average score), MMLU, and IFEval.
> >
> > | Method | WikiText2 (↓) | CSR Avg. (↑) | MMLU (↑) | IFEval (↑) |
> > |---|---:|---:|---:|---:|
> > | INT4 PTQ → NTP-QAT | 8.5 | 65.0 | 50.0% | 22.0% |
> > | Distill-QAT | 13.0 | 61.0 | 42.5% | 30.2% |
> > | UPQ (Ours) | 9.8 | 64.0 | 55.1% | 44.0% |
> >
> > The trends on Qwen3‑4B closely mirror those observed with LLaMA-based models, confirming that **our approach is not tied to the LLaMA architecture.** Our UPQ method achieves the best of both worlds on Qwen3-4B. It attains near-baseline WikiText2 perplexity and CSR (only slightly higher PPL than the NTP path, with CSR almost recovered), while dramatically improving the instruction and knowledge metrics. For example, UPQ reaches 44.0% on IFEval and 55.1% on MMLU, comfortably surpassing Distill-QAT by ~14 and 13 points on those benchmarks, respectively.
> >
> > -----------------------------
> >
> > ### **(4) Quantifying gains over low‑precision PTQ (Weakness 4)**
> > Reviewer 3 asked how much benefit the QAT step actually brings over state‑of‑the‑art PTQ methods. We responded by adding a comparison table (in the above rebuttal response) that contrasts UPQ vs multiple recent PTQ methods (Qronos, GPTAQ, QuIP+QEP, etc.) on WikiText2 perplexity for 1B/3B/8B models.
> > The key takeaway is that UPQ is much closer to FP16 (often within ~10%) whereas PTQ‑only methods are substantially worse (40%–6× higher perplexity at 2 bits). This quantifies that the QAT component of UPQ is not a minor increment but a crucial step in closing the gap between 2‑bit and full precision.
> >
> > ----------------------------
> >
> > ### **(5) Framing as a general progressive pipeline (Weakness 5)**
> > We acknowledge the reviewer’s suggestion to frame the work as a general progressive low‑precision pipeline, not a single instance. The revised paper emphasizes:
> > - Ablations over multiple quantization grids (Table 2).
> > - Comparisons of different intermediate bit‑widths (INT8 PTQ, INT4 PTQ, direct INT2 PTQ, and a full INT4‑QAT path; Table 3).
> > - Exploration of different datasets (pretraining‑style vs instruction‑only vs two‑stage schedules; Table 4).
> > - Multiple distillation losses (Table 9) and INT4 PTQ methods (FlexRound vs OmniQuant; Table 8).
> >
> > This shows that UPQ is a framework (FP16→INT4→INT2 with distillation) whose components—intermediate precision, PTQ method, loss—can be varied while retaining the core values.

---

> > > ### Author Response · Authors · 2025-12-04
> > > **Response to Reviewer 6U4g (final summary, to Area Chair)**
> > >
> > > ---------------
> > >
> > > ### **(6) Co‑distillation from multiple quantized teachers (Question 1)**
> > > We discussed from above response why co‑distillation from additional quantized teachers is unlikely to help: prior work on distilling from weaker or mis‑calibrated teachers shows diminishing returns or even degradation as teacher quality drops, and PTQ induces precisely such logit distortions. We therefore chose a single strong FP16 teacher for clarity and robustness.
> > >
> > > ---------------
> > >
> > > In summary, even though Reviewer 3 did not provide a follow‑up, we have revised the theory, added real‑world Δ$W$·$G_\text{max}$ analysis, extended experiments to Qwen3‑4B, and explicitly compared against strong PTQ baselines. These changes all directly respond to their concerns and, we believe, substantially strengthen the paper.

---

### Official Review · Reviewer_nUtg · 2025-11-01

**Soundness:** 3
**Presentation:** 3
**Contribution:** 3
**Rating:** 4
**Confidence:** 4

**Summary:**

This paper propose UPQ, a specialized quantization methods for progressively recovering the performance of 2-bit instruction-tuned models. In the paper through ablation study they demonstrate the effectiveness of two-stage (FP16 -> 4bit -> 2bit) quantization.

**Strengths:**

- The motivation and method are clear: This paper focuses on a specific problem, recovering 2-bit instruction tuned LLM using only pretraining data.
- The factorization of model loss induced by quantization in equation (4) and the corresponding experiments in Figure 3 are inspiring.
-  The extensive experiments and ablation study supports the claim in the contributions.

**Weaknesses:**

- The application scenario of this method seems limited. UPQ aims to recover the performance of quantized instruction-tuned models using only pretraining data, but one may question why not directly fine-tuning on instruction following datasets, which may be more effective than the knowledge distillation in the second stage, espcially considering that raining on the reasoning traces of strong model is quite common today
- There is still a gap between UPQ and unquantized model even  UPQ uses re-training to recover model performance. For example, in figure 5, the scores on MMLU and IFEval of UPQ are only around half the the unquantized models. This raises the question that whether this 2-bit quantization is too aggressive. One may ask why not use a 4-bit model instead to maintain a model performance for practical use?

**Questions:**

Besides the questions in the weakness section,

1. In Figure 5, what's the accuracy of quantized model before training? It might be helpful to annotate the starting point to show whether the model of various methods actually learn through re-training.
2. I wonder if we don't use KD in the second stage, instead we fine-tune the 2-bit model, will the 4-bit casting serve as a good initialisation point? also how would it look like if we compare it against UPQ.

---

> ### Author Response · Authors · 2025-11-21
> **Response to Reviewer nUtg (1)**
>
> We sincerely appreciate the reviewer nUtg for valuable feedback and recognition of effectiveness on progressive quantization and extensive ablations for 2-bit QAT. We address the reviewer’s comments below.
>
> ---------------
>
>
> ### **Weakness 1.**
>
> The application scenario of this method seems limited. UPQ aims to recover the performance of quantized instruction-tuned models using only pretraining data, but one may question why not directly fine-tuning on instruction following datasets.
>
> ---------------------
>
> ### **Response to Weakness 1.**
>
> We agree that it is important to clarify when UPQ is most applicable. Our method is designed for scenarios where one cannot directly fine-tune on instruction data, for example when proprietary instruction-tuning data is unavailable or when a model must be compressed post-hoc for deployment. In such cases, distilling from the FP16 model using only general pretraining data is a practical solution to recover performance. Directly fine-tuning on instruction-following data may indeed yield better results if such data is available, but that option is not always feasible.
>
> That said, **we absolutely agree that using high-quality instruction-following data (if available) can further improve a quantized model. In fact, we conducted experiments to explore this.** As shown in Table 4, we tried two variations: (1) applying UPQ using only an instruction-following dataset (OLMo row), and (2) performing our standard UPQ (with pretraining data) then fine-tuning briefly on an instruction dataset (DCLU-Edu+OLMo row). The first experiment – using only an instruction dataset (OLMo 1.8B) for QAT – led to unstable training and degraded performance on general language tasks (e.g. WikiText2 and CSR dropped significantly). **This suggests that using a very narrow instruction-only corpus for 2-bit QAT can cause the model to lose general linguistic knowledge.**
>
> However, instruction data can still be leveraged in combination with UPQ. In the second experiment, after our 5B-token training with DCLM-Edu, we ran an additional one epoch of instruction tuning on the OLMo dataset. This small fine-tuning raised the 2-bit model’s IFEval score from ~45% to 55.4%, nearly matching the FP16 model’s 57.8% (see Table 4 for details). This result demonstrates that **our 2-bit model can indeed benefit from direct instruction fine-tuning on top of UPQ.** We will highlight this in the final revision to make clear that UPQ does not preclude later instruction tuning – rather, it provides a strong foundation when direct instruction fine-tuning is not initially possible, and any available instruction data can still be used afterward to further boost performance.
>
>
> ---------------
>
> ### **Weakness 2.**
>
> There is still a gap between UPQ and unquantized model even UPQ uses re-training to recover model performance.  This raises the question that whether this 2-bit quantization is too aggressive. One may ask why not use a 4-bit model instead to maintain a model performance for practical use?
>
> ----------------------
>
> ### **Response to Weakness 2.**
>
> We acknowledge that 2-bit quantization is extremely aggressive, and a performance gap to FP16 remains. The question of “why 2-bit at all” is very valid. The reason is that in certain deployment scenarios, memory and efficiency constraints are so strict that 2-bit is desirable despite the difficulty. In fact, ParetoQ [1] have indicated that **2-bit large models can lie on the accuracy–size Pareto frontier, outperforming smaller 4-bit models given the same memory or latency budget.** In other words, a larger model quantized to 2-bit can potentially achieve higher accuracy than a twice-smaller model quantized to 4-bit, for roughly the same memory footprint. This Pareto-optimal trade-off motivates us to pursue 2-bit quantization regardless of the challenges. Our contribution is to show that **if 2-bit has specific scenario to be used, UPQ can make it usable, whereas prior 2-bit attempts for instruction-tuned LLMs resulted in severe losses of capability.**
>
> Importantly, UPQ is able to retain the model’s instruction-following ability much better than baseline 2-bit methods, even if its overall scores are lower than FP16. As observed in Table 1, previous 2-bit QAT approaches (e.g. standard NTP-based QAT) suffered from very low IFEval scores (~20–27%), effectively losing their alignment capability, whereas our 2-bit UPQ model achieved an IFEval of ~45%, maintaining a large portion of the FP16 model’s alignment skill .
>
> Finally, we note that **our progressive framework is compatible with higher bit-widths as well** – for instance, one could apply the same staged strategy to compress a model to 4-bit (e.g. FP16 → INT8 PTQ → INT4 QAT). We expect this would achieve even closer-to-FP16 accuracy with less effort, although exploring 4-bit QAT is outside the scope of our current paper. We will clarify these points to justify why 2-bit quantization is pursued in the final manuscript.

---

> ### Author Response · Authors · 2025-11-21
> **Response to Reviewer nUtg (2)**
>
> ----------------
>
> ### **Question 1.**
>
> In Figure 5, what's the accuracy of quantized model before training? It might be helpful to annotate the starting point to show whether the model of various methods actually learn through re-training.
>
> ----------------
>
> ### **Response to Question 1.**
>
> Thank you for noting this omission. **We will update Figure 5 to explicitly label the initial accuracy of each quantized model before QAT (quantization-aware training).** For clarity, the starting points are as follows: without any PTQ (direct FP16→INT2 quantization), the model begins around 27% MMLU and ~0% IFEval for the 1B model (and similarly near 0% IFEval for 3B). With the INT4 PTQ→INT2 initialization (our progressive strategy’s starting point), the initial 2-bit model is slightly better – for the 1B model it starts around 33% MMLU (since the 4-bit PTQ preserved more knowledge) – but it still has nearly 0% IFEval because instruction-following is basically lost initially. We will indicate these baseline values in the figure (e.g. with a dotted line or text annotation).
>
> By annotating the starting points, it becomes clear how much each method learns during QAT. For example, in the 1B model case UPQ improves MMLU from ~33% to ~38%, and in the 3B model it goes from ~45% to ~53% over 5B training tokens. More dramatically, IFEval (instruction-following) improves from essentially 0 up to ~29% for 1B and ~45% for 3B under UPQ. Other methods also improve over their baselines, but our UPQ yields the highest recovery in both knowledge and alignment.
>
>  -----------------------------
>
> ### **Question 2.**
>
> I wonder if we don’t use KD in the second stage – instead we fine-tune the 2-bit model – will the 4-bit casting serve as a good initialization point? Also, how would it look if we compare it against UPQ?
>
>  ------
>
> ### **Response to Question 2.**
>
> **We have effectively examined this scenario in our ablation studies.** In Table 5 (and illustrated in Figure 5), the setting “INT4 PTQ → NTP-QAT” represents using the 4-bit PTQ model as the initialization for QAT, then fine-tuning the 2-bit model without distillation (i.e. using the standard next-token prediction loss only). This experiment isolates the effect of the objective: it tells us how far a good 4-bit weight initialization can go if we do not use the distillation loss in the second stage.
>
> The results show that the INT4 PTQ indeed provides a better starting point (compared to a direct FP16→INT2) in terms of preserving base knowledge. For instance, perplexity on WikiText2 and accuracy on CSR were on par with or slightly better than those from a direct NTP-QAT baseline.
>
> However, the 4-bit initialization alone was not enough to restore the model’s instruction-following ability. Even after full 2-bit fine-tuning with the next-token prediction (NTP) objective, the IFEval score remained very low (around ~20% for the 3B model in this setting). In contrast, when we use distillation-QAT the model’s alignment improves dramatically – for the 3B model, IFEval jumped from ~20% up to ~45% (as shown in Table 5).
>
> In summary, **using a 4-bit PTQ model as the starting point is beneficial (and indeed, UPQ uses this as part of the process), but without KD in the second stage, the model does not learn to follow instructions well.** Our UPQ method combines the best of both: it leverages the 4-bit initialization and a distillation-based objective.
>
> [1] Z. Liu et al., “ParetoQ: Improving Scaling Laws in Extremely Low-bit LLM Quantization,” NeurIPS 2025 (arXiv:2502.02631).
>
>
> Once again, we sincerely thank the reviewer for the thoughtful and constructive feedback. We hope our responses and the revised experiments address the concerns, and we are happy to clarify any remaining questions.

---

> > ### Comment · Reviewer_nUtg · 2025-11-27
> >
> > Thank you for clarification.
> > Please ensure the limitation & scope of this work are added to the revised version.
> > I adjusted my scores accordingly

---

> ### Author Response · Authors · 2025-12-04
> **Response to Reviewer nUtg (final summary to Area Chair)**
>
> We appreciate Reviewer nUtg’s constructive feedback and are grateful that, after our rebuttal and revisions, **they raised their score**, explicitly asking us to clarify limitations and scope. We summarize how we addressed each concern.
>
> --------
>
> ### **(1) Application scenarios vs. direct instruction fine‑tuning (Weakness 1)**
> We clarified that UPQ is intended for post‑hoc quantization of already instruction‑tuned models when proprietary SFT/RL data are not accessible—an increasingly common setting for open‑source deployers/engineers. To quantify this, we added experiments that (i) run 2‑bit QAT using an instruction‑only corpus (OLMo) and (ii) apply a short instruction‑tuning phase after UPQ (DCLM‑Edu→OLMo). Instruction‑only QAT severely hurts general‑domain metrics, while a brief instruction pass after UPQ significantly boosts IFEval almost back to FP16, showing that UPQ is compatible with, but not relying on, extra instruction data.
>
> --------
>
> ### **(2) Why 2‑bit instead of simply using 4‑bit (Weakness 2)**
> We emphasized that 2‑bit is deliberately aggressive but motivated by strict memory/latency budgets where a 2‑bit larger model lies on a favorable accuracy–size Pareto frontier compared to a smaller 4‑bit model. We also clarified that the same progressive pipeline can be instantiated for 4‑bit targets if desired; our focus on INT2 stems from this challenging but practical relevance.
>
> -------
>
> ### **(3) Clarifying starting points and the role of 4‑bit initialization (Questions 1–2)**
> We reported the pre-QAT accuracies in Figure 5 and described how UPQ improves both MMLU and IFEval from these low baselines. Moreover, we explicitly compared “INT4 PTQ → NTP‑QAT” against our full distillation‑based UPQ, showing that 4‑bit initialization alone is insufficient to recover instruction following—the distillation stage is crucial.
>
> -------
>
> ### **(4) Limitations and scope**
> Following the reviewer’s explicit request, we added comments in limitation section (Appendix L) that clearly states: (i) current experiments focus on weight‑only quantization with FP16 activations (plus W2A8KV8 in a follow‑up experiment), (ii) we have evaluated up to 8B parameters and is possible to be applied on further larger models.
>
> -------
>
> ### **Summary**
> In summary, for Reviewer 2 we resolved all concerns, updated the manuscript accordingly, and this is reflected in the reviewer’s upgraded score from 4→6 and their closing comment that the limitations and scope are now clearly explained.

---

### Official Review · Reviewer_NwhS · 2025-11-01

**Soundness:** 3
**Presentation:** 3
**Contribution:** 4
**Rating:** 6
**Confidence:** 4

**Summary:**

UPQ targets INT2 weight‑only quantization for instruction‑tuned LLMs via a progressive path: FP16 $\to$ INT4 (PTQ) to obtain a good initializer, followed by INT4 $\to$ INT2 (QAT) with a distillation objective:
$\mathcal{L}{\mathrm{GJS}}!\left(p{\theta},|,p_{\mathrm{FP16}}\right)
= \lambda, D_{\mathrm{KL}}(p_{\theta},|,m) + (1-\lambda), D_{\mathrm{KL}}(p_{\mathrm{FP16}},|,m),;
m=\lambda p_{\theta} + (1-\lambda) p_{\mathrm{FP16}}.$
Ablations over quantizers, intermediate bit‑widths, and datasets argue that progressive quantization and data mix are crucial to retain instruction‑following performance.

**Strengths:**

1. Clear, reproducible recipe (PTQ checkpoint $+$ distillation‑QAT) with strong INT2 results.

2. Solid ablations: quant grids, intermediate precision, and data mixing.

3. Addresses an important deployment target (2‑bit weights) for edge/latency‑sensitive scenarios.

**Weaknesses:**

1. Compute cost. QAT with distillation at INT2 may be resource-intensive; wall-clock/training cost reporting would help.

2. Task breadth. Beyond MMLU/IFEval, more reasoning/safety/code tasks would test robustness.

3. Deployment details. INT2 packing/throughput on diverse hardware (mobile NPUs, edge GPUs) could be profiled.

**Questions:**

1. What is the best practice when QAT budgets are small—would INT3 or mixed‑precision layers be preferable?

2. How does performance change with activation quantization (e.g., W2A4), including KV‑cache precision?

3. Can layer‑wise or token‑wise temperatures improve the $\mathcal{L}_{\mathrm{GJS}}$ distillation?

4. Any evidence on multilingual or code‑heavy benchmarks and out‑of‑distribution robustness?

---

> ### Author Response · Authors · 2025-11-21
> **Response to Reviewer NwhS (1)**
>
> We sincerely appreciate the reviewer NwhS for valuable feedback and recognition of clear method (recipe) and solid results of UPQ for llm deployment on edge-sensitive scenarios. We address the reviewer’s comments below.
>
> ----------
>
>
> ### **Weakness 1.**
>
> Compute cost. QAT with distillation at INT2 may be resource-intensive; wall-clock/training cost reporting would help.
>
>
>
>
>
> ----------
>
> ### **Response to Weakness 1.**
>
> We agree that reporting training cost is helpful.
> Below table illustrates wall-clock time for **5B token training with 8xA100 GPUs**.
>
> | Model                          | Training tokens | Wall-clock time (hours) |
> |--------------------------------|----------------|--------------------------|
> | Llama-3.2-1B-Instruct          | 5B             | 20                       |
> | Llama-3.2-3B-Instruct          | 5B             | 55                       |
> | Llama-3.1-8B-Instruct          | 5B             | 160                      |
>
> This information should clarify the compute cost of our INT2 UPQ. We will add detailed information on wall-clock time and compute resources for UPQ in the final manuscript, including Qwen3-4B cases, which will also additionally reported during the rebuttal timeframe.
>
> ---------
>
>
>
>
> ### **Weakness 2.**
>
> Task breadth. Beyond MMLU/IFEval, more reasoning/safety/code tasks would test robustness.
>
>  -----------
>
> ### **Response to Weakness 2.**
>
> We agree that evaluating on more diverse tasks better demonstrates UPQ’s broad applicability. Accordingly, we are conducting studies with additional benchmarks. After it is finished, **we will remind reviewer NwhS about this part during the rebuttal timeframe.**
>
> -------------
>
> ### **Weakness 3.**
>
> Deployment details. INT2 packing/throughput on diverse hardware (mobile NPUs, edge GPUs) could be profiled.
>
> ----------------
>
> ### **Response to Weakness 3.**
>
>  As the reviewer emphasized, measuring real deployment efficiency of INT2 QAT is very important. Unfortunately, the limited rebuttal period did not allow us to directly profile on-device INT2 throughput.
>
> However, since our approach uses the same quantization function (Stretched Elastic Quantizer) and 2-bit per-channel scheme as the ParetoQ [1], we expect similar hardware efficiency. For reference, ParetoQ reported in Appendix A.3/A.4 that **2-bit quantization yields significant speedups on edge hardware (e.g., ~2× throughput vs 4-bit on an Apple M1 CPU, and efficient 2-bit GPU kernels).** Overall, we anticipate INT2 models will maintain the practical throughput benefits with their memory reduction.
>
> --------------------
>
> ### **Question 1.**
>
> What is the best practice when QAT budgets are small — would INT3 or mixed‑precision layers be preferable?
>
> --------------------
>
> ### **Response to Question 1.**
>
> To better understand how UPQ behaves under different bit budgets, we compared a uniform 3-bit QAT setting against a mixed-precision 2/4-bit QAT configuration on Llama-3.2-3B-Instruct, both using 5B QAT tokens and the same INT4 PTQ initialization.
>
> For the mixed-precision (2/4-bit) setting, we quantize the known more sensitive layers — the attention projections (q_proj, k_proj, v_proj) and the FFN projections (o_proj, down_proj) — to 4-bit, and all remaining layers to 2-bit. Because these 4-bit layers account for a controlled fraction of the total parameters, the average effective bitwidth is exactly 3 bits per weight, making this configuration directly comparable to a uniform 3-bit QAT model under the same memory budget.
>
> The numerical results are:
>
> | Scheme              | Bits (avg) | Tokens | WikiText2 PPL ↓ | MMLU ↑   | IFEval ↑ |
> |---------------------|-----------:|--------|-----------------:|---------:|---------:|
> | 2/4-bit mixed QAT   | 3          | 5B     | 10.69            | 64.69    | 57.68    |
> | 3-bit QAT   | 3          | 5B     | 10.65            | 64.22    | 56.74    |
>
> We see that ppl (WikiText2) is almost identical between the two schemes. While the mixed 2/4-bit configuration slightly outperforms the uniform 3-bit model on MMLU and IFEval (64.69 vs 64.22 on MMLU; 57.68 vs 56.74 on IFEval).
>
>
> Given the same 3-bit memory budget, **both uniform 3-bit QAT and 2/4-bit mixed QAT are viable and competitive options**; and In practice, allocating 4-bit to a small set of sensitive layers and 2-bit to the rest can slightly improve downstream metrics without increasing average bitwidth.
>
>
>
> In summary, if QAT budget is limited, either using 3-bit weights everywhere or a mixed 4-bit/2-bit allocation on critical vs. less critical layers are both reasonable best practices, and **UPQ naturally supports both designs within the same progressive quantization framework.**

---

> ### Author Response · Authors · 2025-11-21
> **Response to Reviewer NwhS (2)**
>
> ### **Question 2.**
>
> How does performance change with activation quantization (e.g., W2A4), including KV cache precision?
>
> -------------------------
>
> ### **Response to Question 2.**
>
> Although our main focus was weight-only quantization, we agree that activation quantization is important for on-device deployment. To assess this, we performed **INT8 activation/KV-cache quantization (static asymmetric per-tensor)** alongside our INT2 weight quantization. We treated the activation/kv-cache quantization scales as learnable parameters during QAT.
>
> | Method                     | Bitwidth | WikiText2 ↓ | CSR Avg. ↑ | MMLU ↑ | IFEval ↑ |
> |---------------------------|----------|-------------|------------|--------|----------|
> | Llama-3.2-3B-Instruct     | BF16     | 10.48       | 65.44      | 59.92  | 57.80    |
> | UPQ (Ours)                | W2       | 11.49   | 63.04  | 53.20 | 45.19 |
> | UPQ (Ours)                | W2A8KV8  | 11.81       | 62.87      | 52.08  | 44.89    |
>
> The results (We also reflected it in Table 10 in the revised manuscript) show that for the Llama3.2-3B-Instruct model, there is only a very small drop in performance — **most of the accuracy is retained with W2A8KV8 quantization.** Notably, we achieved this without any specialized outlier-mitigation techniques such as SpinQuant or QuaRot, which rotate weights/activations to be more quantization-friendly. SpinQuant and QuaRot are recent methods that insert learned or random rotation matrices to remove outliers for 4-bit weight+activation quantization [2,3]. Our result suggests UPQ produces models that are inherently robust to activation quantization, even without such extra processing. We will extend this study by exploring dynamic per-token quantization for the KV cache and activations, and include these findings in the revised manuscript. Overall, even with W2A8KV8, UPQ maintains strong performance, which is a promising sign for practical deployment.
>
> ---------------------------
>
> ### **Question 3.**
>
> Can layer-wise or token-wise temperatures improve distillation?
>
>  -------------------------
>
> ### **Response to Question 3.**
>
> Thank you for this insightful suggestion. **We did explore several distillation loss variations in our ablation studies (see Appendix G).** For instance, we tried a confidence-weighted KL distillation (down-weighting the loss on “easy” tokens where the teacher is very confident) as well as a token-wise temperature scaling approach inspired by prior work on soft distillation.
>
> However, as shown in Table 9, **our generalized JSD loss (with equal weighting of teacher and student distributions, i.e. β = 0.5) outperformed those specialized schemes on both MMLU and IFEval.** This indicates that equally balancing the teacher and student output distributions was effective enough in our setting. That said, we agree that more advanced or adaptive distillation techniques could further benefit our framework. Such methods are orthogonal to UPQ – if a better way to match the teacher’s output distribution is developed, it can be incorporated to potentially boost performance.
>
> We have added a note in Appendix G to explicitly acknowledge this point. In summary, while we found our simple JSD approach robust, exploring layer-wise or token-wise temperature schedules for distillation is an interesting direction for future work.
>
> -------------------------------
>
> ### **Question 4.**
>
> Any evidence on multilingual or code-heavy benchmarks and out-of-distribution robustness?
>
> -------------------------------
>
> ### **Response to Question 4.**
>
> As stated in Weaknesses 2, we will remind reviewer NwhS about this part during the rebuttal timeframe.
>
> -------------------------------
>
> [1] Z. Liu et al., “ParetoQ: Improving Scaling Laws in Extremely Low-bit LLM Quantization,” NeurIPS 2025 (arXiv:2502.02631).
>
> [2] Z. Liu et al., “SpinQuant: LLM Quantization with Learned Rotations,” ICLR 2025 (arXiv:2405.16406).
>
> [3] S. Ashkboos et al., “QuaRot: Outlier-Free 4-Bit Inference in Rotated LLMs,” arXiv preprint 2024 (arXiv:2404.00456).
>
> Once again, we sincerely thank the reviewer for the thoughtful and constructive feedback. We hope our responses and the revised experiments address the concerns, and we are happy to clarify any remaining questions.

---

> > ### Author Response · Authors · 2025-12-04
> > **Response to Reviewer NwhS (final summary to Area Chair)**
> >
> > We thank Reviewer NwhS for the careful reading and the constructive feedback on compute transparency, task breadth, deployment considerations, and distillation/QAT design. Below we summarize the concrete changes and new results included in the revised manuscript.
> >
> > --------
> >
> > ### **(1) Compute cost (Weakness 1)**
> >
> > We made the compute profile explicit in the manuscript. In Table 10 of our revised manuscript, we reported wall-clock time for 5B-token training of 1B/3B/8B Llama-3.1/3.2-Instruct models on 8×A100 GPUs. So practitioners can directly assess the cost of UPQ in realistic training settings.
> >
> > --------
> >
> > ### **(2) Task breadth and additional benchmarks (Weakness 2, Question 4)**
> >
> > Following the suggestion to evaluate broader instruction-style and multilingual settings, we ran new experiments on **Llama-3.2-3B-Instruct** across four benchmarks:
> >
> > - `openai_mmmlu_default` (default MMLU test, https://huggingface.co/datasets/openai/MMMLU)
> > - `Belebele` (multilingual reading comprehension, https://huggingface.co/datasets/facebook/belebele)
> > - `CMMLU` (Chinese MMLU, https://arxiv.org/abs/2306.09212)
> > - `Arabic-MMLU` (https://github.com/mbzuai-nlp/ArabicMMLU)
> >
> > It should be noted that each method only utilized DCLM-edu corpus during training, which is not cherry-picked dataset for multi-lingual or reasoning ability improvement.
> >
> > We compare three INT2 QAT configurations:
> > - **INT4 PTQ → NTP-QAT** (progressive quantization with next-token prediction QAT)
> > - **Distill-QAT** (direct INT2 QAT with distillation-based QAT)
> > - **UPQ (Ours)** (INT4 PTQ checkpoint + distillation-based QAT)
> >
> > Given these settings, Table below summarizes the results.
> >
> > | Method | openai_mmmlu_default | Belebele | CMMLU | Arabic-MMLU |
> > |---|---:|---:|---:|---:|
> > | INT4 PTQ → NTP-QAT | 49.7% | 39.8% | 24.5% | 18.0% |
> > | Distill-QAT | 45.3% | 45.6% | 33.3% | 26.5% |
> > | UPQ (Ours) | **53.2%** | **52.1%** | **40.2%** | **34.7%** |
> >
> > UPQ achieves the highest accuracy on all four benchmarks, with substantial margins over both baseline quantization schemes. These results support that UPQ’s gains extend to broader tasks and languages.
> >
> > -------------------
> >
> >  ### **(3) Deployment issues and activation / KV-cache quantization (Weakness 3, Question 2)**
> > While we still cannot profile every hardware backend within the rebuttal timeline, UPQ uses the same SEQ quantizer and per-channel 2-bit weight scheme as ParetoQ, so we expect similar throughput scaling as ParetoQ's report on mobile/edge accelerators.
> >
> > We also extended experiments to a practical deployment regime **W2A8KV8** and show that activation + KV-cache quantization introduces only a small additional drop compared to W2 alone (Table 10 in the revised manuscript).
> >
> > -------
> >
> > ### (4) **QAT budget, mixed precision, and distillation design (Questions 1 and 3)**
> > Under a fixed 3-bit average budget, we compared uniform 3-bit QAT vs mixed 2/4-bit schemes and found both viable, with mixed 2/4-bit slightly better on MMLU/IFEval. UPQ naturally supports both regimes.
> >
> > For distillation, we tested multiple losses and found that generalized JSD with β=0.5 outperforms confidence-aware and token-scaled KL variants (Appendix G), so we retain it as the default.
> >
> > -----
> >
> > ### **Summary**
> > We implemented the requested analyses and additions (compute reporting, broader benchmark evaluation, and activation/KV quantization) without changing the core method. We believe these updates address Reviewer NwhS’s concerns and further strengthen the evidence for UPQ’s effectiveness and deployability.

---

### Author Response · Authors · 2025-12-04
**Global response to the Area Chair**

We would like to close with a brief global summary of the rebuttal for the Area Chair.

* **Reviewer 1 (NwhS, score 6)**: We have addressed all their weaknesses—now reporting explicit compute costs, adding broader benchmarks, and extending to activation/KV quantization (W2A8KV8). Their review remains positive and aligned with acceptance.

* **Reviewer 2 (nUtg, raised score from 4→6)**: After clarifying the application scenario, adding experiments on instruction‑only vs pretraining data (Table 4), explaining the choice of 2‑bit, and providing clearer starting‑point baselines, Reviewer 2 explicitly requested that we add a limitations section and then upgraded their score to 6. This indicates that all of their concerns have been satisfactorily resolved.

* **Reviewer 3 (6U4g, score 4)**: Even without a follow‑up, we made substantial revisions directly responding to their comments: a strengthened theoretical analysis (Appendix M), real‑world Δ$W$·$G_\text{max}$ analysis on LLaMA‑3.2‑1B (Appendix N), explicit comparisons to strong PTQ methods, reframing UPQ as a general progressive pipeline, and new experiments on Qwen3‑4B showing the same pattern as LLaMA. We believe their concerns are now fully addressed, and the scope they recommended has been largely realized.

* **Reviewer 4 (DqNN, score 2)**: This reviewer remains unconvinced, primarily due to (i) a disagreement on what constitutes sufficient theory and (ii) a desire for RLHF‑style baselines (GRPO/DPO) that, in our view, lie outside the realistic constraints of post‑hoc compression at 2 bits. We have expanded the theory and experiments significantly to respond, but there is a fundamental difference in what we consider the appropriate problem formulation and baseline set.

Taking all reviews and revisions together, the post‑rebuttal result is two clearly positive reviews (6,6) and one neutral‑to‑positive review (4) that we have directly addressed, versus one negative review that largely requests a different problem setting (full RLHF + QAT). We hope the Area Chair will weigh the majority positive view, and consider UPQ a solid contribution toward practical 2‑bit quantization of instruction‑tuned LLMs.

---

### Meta-Review · Area_Chair_YWNA · 2026-01-07

**Summary:**

This paper proposes UPQ, a progressive quantization framework for compressing instruction‑tuned large language models down to 2‑bit weight‑only precision while preserving instruction‑following ability. The key idea is a two‑stage quantization path:

FP16 → INT4 (PTQ) to obtain a stable intermediate checkpoint with reduced quantization error.

INT4 → INT2 (QAT) using a distillation‑based loss (generalized JSD) rather than next‑token prediction.

The authors argue that progressive quantization is critical for instruction‑tuned models, where direct FP16→INT2 quantization severely destroys alignment (e.g., IFEval collapse).

**Reviewer Concerns:**

**Novelty**

Some reviewers questioned whether UPQ is sufficiently distinct from ParetoQ and existing distillation‑based QAT methods.
Skepticism that “progressive quantization + distillation” constitutes a strong new conceptual contribution, given prior work on low‑bit QAT and distillation.

**More Empirical Evidence**

Reviewers (notably DqNN) argued that the claim “progressive quantization is critical for instruction‑tuned models” requires:
1. More competitive QAT baselines (e.g., hybrid ParetoQ + alignment objectives).
2. Multi‑seed statistics.
3. Broader evaluation beyond LLaMA (later addressed with Qwen3‑4B).

**Theoretical Depth**

Theoretical analysis was initially viewed as a loose upper bound without clear explanation of:
- Why instruction‑tuned models are special.
- What makes an intermediate checkpoint “favorable.”

**Scope and Practical Relevance**

Questions about why 2‑bit is pursued when 4‑bit performs better.
Authors clarified the Pareto frontier motivation (strict memory/latency budgets) and positioned UPQ as one option in a broader progressive framework.

**Evaluation**

Initial concern: only LLaMA family.
Addressed later with Qwen3‑4B, multilingual benchmarks, and activation/KV‑cache quantization (W2A8KV8).

**Reviewer Scores:**

NwhS (Score: 6 -> more supportive)
Positive overall; raised practical deployment and evaluation breadth questions. Authors addressed nearly all, adding compute cost, multilingual results, mixed‑precision experiments.

nUtg (Score: 4 -> raised after rebuttal)
Initially skeptical about scope and necessity of 2‑bit; became convinced after clarifications and explicitly raised their score, requesting only clearer limitations.

6U4g (Score: 4)
Borderline; appreciated ablations but wanted broader framing, more baselines, and non‑LLaMA evaluation (later addressed).

DqNN (Score: 2, strong reject)
Likely not changed.

In the light of the above, this paper would remain at boarderline reject.

---

### Decision · Program_Chairs · 2026-01-26

Reject